# Data-Centric Learning from Unlabeled Graphs with Diffusion Model

**Gang Liu**
University of Notre Dame
gliu7@nd.edu

**Eric Inae**
University of Notre Dame
einae@nd.edu

**Tong Zhao**
Snap Inc.
tzhao@snap.com

**Jiaxin Xu**
University of Notre Dame
jxu24@nd.edu

**Tengfei Luo**
University of Notre Dame
tluo@nd.edu

**Meng Jiang**
University of Notre Dame
mjiang2@nd.edu

## Abstract

Graph property prediction tasks are important and numerous. While each task offers a small size of labeled examples, unlabeled graphs have been collected from various sources and at a large scale. A conventional approach is training a model with the unlabeled graphs on self-supervised tasks and then fine-tuning the model on the prediction tasks. However, the self-supervised task knowledge could not be aligned or sometimes conflicted with what the predictions needed. In this paper, we propose to extract the knowledge underlying the large set of unlabeled graphs as a specific set of useful data points to augment each property prediction model. We use a diffusion model to fully utilize the unlabeled graphs and design two new objectives to guide the model's denoising process with each task's labeled data to generate task-specific graph examples and their labels. Experiments demonstrate that our data-centric approach performs significantly better than fifteen existing various methods on fifteen tasks. The performance improvement brought by unlabeled data is *visible* as the generated labeled examples unlike the self-supervised learning.

## 1 Introduction

Graph data such as molecules and polymers are found to have attractive properties in drug and material discovery (Böhm et al., 2004; Huang et al., 2021), but annotating them requires specialized knowledge, as well as lengthy and costly experiments in wet labs (Cormack and Elorza, 2004). So, it is important for graph property predictors to learn *useful knowledge* from unlabeled graphs.

Self-supervised learning (Hu et al., 2019; Rong et al., 2020; You et al., 2021; Kim et al., 2022) utilizes unlabeled graphs to learn through *predictive tasks* or *contrastive tasks* to represent and transfer the knowledge as *model parameters*. Despite the empirical success in language and vision (Brown et al., 2020; He et al., 2022), their performance on graph data applications remains unsatisfactory because of the significant gap between the graph self-supervised task and the graph label prediction task. Models trained on node attribute prediction (Hu et al., 2019) as a simple *predictive* self-supervised task extract too limited knowledge from the graph structure, which has been observed after too fast convergence (Sun et al., 2022). More complex tasks like graph context prediction (Hu et al., 2019; Zhang et al., 2021) may transfer knowledge that conflicts with downstream tasks. Aromatic rings, for instance, are a prevalent structure in molecules (Maziarka et al., 2020) and are considered valuable in context prediction tasks (Zhang et al., 2021). However, graph properties such as oxygen permeability can be more related to non-aromatic rings in some cases (Liu et al., 2022a), which is overlooked if not using tailored predictive tasks specifically for downstream tasks. As predictive tasks strive for

37th Conference on Neural Information Processing Systems (NeurIPS 2023).

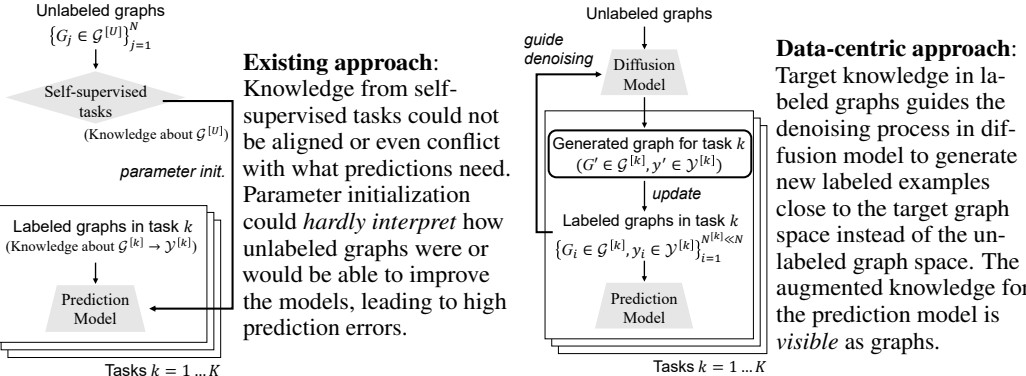

Figure 1: Comparing the diagrams of the existing approach and the proposed approach to learning from unlabeled graphs for a variety of graph property prediction tasks.

universality, the transferred knowledge may force models to focus more on aromatic rings, leading to poor prediction.

On the other line, *contrastive* tasks (You et al., 2021; Kim et al., 2022) aim to learn the similarity between original and perturbed graphs. However, the learned similarity can hardly generalize across tasks (Kim et al., 2022). First, perturbations without domain knowledge, *e.g.,* bioisosteres, do not preserve broad biological properties (Sun et al., 2021). Second, it is difficult, if not impossible, to find universal perturbations that generalize to diverse property prediction tasks. For example, bioisosteric (subgraph) replacements produce similar biological properties for molecules. And they may reduce toxicity (Brown, 2014). So, contrastive tasks with bioisosteric replacement enforce the similarity between toxic and non-toxic molecules. However, models pre-trained on such contrastive tasks hurt the performance on downstream tasks, *e.g.,* toxicity prediction.

Our *data-centric* idea avoids the use of self-supervised tasks that are not appropriate. We use a diffusion probabilistic model (known as *diffusion model*) to capture the data distribution of *unlabeled graphs*, leveraging its capability of distribution coverage, stationarity, and scalability (Dhariwal and Nichol, 2021). At the stage of performing a particular property prediction task, the reverse process, guided by novel task-related optimization objectives, generates new task-specific labeled examples. Minimal sufficient knowledge from the unlabeled data is transferred into these examples, instead of uninterpretable model parameters, and then to enhance the training of prediction models.

To implement our idea, we propose a *Data-Centric Transfer* framework (DCT) based on a diffusion model for graph data, as shown in Figure 1b. It aims to transfer minimal sufficient knowledge from unlabeled graphs to property predictors by data augmentation. The diffusion model gradually adds Gaussian noise to a graph from which a score function (*i.e.,* the gradient of the log probability density) is then learned to estimate the noise step by step to reverse the process. DCT trains the diffusion model on the unlabeled graphs to get ready to augment any labeled dataset. Given a labeled graph from a particular task (*i.e.,* type of property), the diffusion model adds noise to perturb it by a few steps and then generates a new graph through the score function. The new graph could be close to the distribution of the unlabeled graphs for diversity, however, it would lose the relatedness to the target task. So, we add two task-related objectives into the score function to guide the reverse process. When a predictor model $f$ has been trained on the task, given an original labeled graph $G$, the first objective is to optimize the new graph $G'$ to *sufficiently* preserve the label of $G$ with $f$. The second objective is to optimize $G'$ to be very different from (*i.e., minimally* similar to) $G$. These two objectives ensure that $G'$ carries minimal sufficient knowledge from the unlabeled graphs to be an augmentation of $G$. DCT iteratively generates new examples to augment the labeled dataset and progressively trains the prediction model with it.

We test DCT on *fifteen* graph property prediction datasets from three fields: chemistry (molecules), material science (polymers), and biology (protein-protein interaction graphs). DCT achieves the best performance over all these tasks. We find that the state-of-the-art self-supervised methods often struggle to transfer knowledge to regression tasks, etc. DCT reduces the mean absolute error relatively by 13.4% and 10.2% compared to the best baseline on the molecule and polymer graph regression tasks, respectively.

## 2 Problem Definition

Given $K$ property prediction tasks, there are $N^{[k]}$ labeled graph examples for the $k$-th task. They are $\{(G_i, y_i) \mid G_i \in \mathcal{G}^{[k]}, y_i \in \mathcal{Y}^{[k]}\}_{i=1}^{N^{[k]}}$, where $\mathcal{G}^{[k]}$ is the graph space and $\mathcal{Y}^{[k]}$ is the label space of the task. The prediction model with parameters $\theta$ is defined as $f_\theta^{[k]} : \mathcal{G}^{[k]} \to \mathcal{Y}^{[k]}$. $f_\theta^{[k]}$ consists of a GNN and a multi-layer perceptron (MLP). Without the loss of generality, we consider Graph Isomorphism Networks (GIN) (Xu et al., 2019) to encode graph structures. Given a graph $G = (\mathcal{V}, \mathcal{E}) \in \mathcal{G}^{[k]}$ in the task $k$, GIN updates the representation vector of node $v \in \mathcal{V}$ at $l$-layer:

$$\mathbf{h}_v^l = \text{MLP}^l \left( (1 + \epsilon) \cdot \mathbf{h}_v^{l-1} + \sum_{u \in \mathcal{N}(v)} \mathbf{h}_u^{l-1} \right), \tag{1}$$

where $\epsilon$ is a learnable scalar and $u \in \mathcal{N}(v)$ is one of node $v$'s neighbor nodes. After stacking $L$ layers, the READOUT function (*e.g.,* summation) gets the graph representation across all the nodes. The predicted label is:

$$\hat{y} = \text{MLP} \left( \text{READOUT} \left( \{ \mathbf{h}_v^L \mid v \in G \} \right) \right). \tag{2}$$

$f_\theta^{[k]}$ is hard to be well-trained because it is hard to collect graph labels at a large scale ($N^{[k]}$ is small).

Fortunately, regardless of the tasks, a large number of **unlabeled graphs** are usually available from the same or similar domains. Self-supervised learning methods (Hu et al., 2019) rely on hand-crafted tasks to extract *knowledge* from the unlabeled examples $\{G_j \in \mathcal{G}^{[U]}, j = 1, \ldots, N\}$ as *pre-trained model parameters $\theta$*. The uninterpretable parameters are transferred to warm up the prediction models $\{f_\theta^{[k]}\}_{k=1}^K$ on the $K$ downstream graph property prediction tasks. However, the gap and even conflict between the self-supervised tasks and the property prediction tasks lead to suboptimal performance of the prediction models. In the next section, we present the DCT framework that transfers knowledge from the unlabeled graphs with a data-centric approach.

## 3 The Data-Centric Transfer Framework

### 3.1 Overview of Proposed Framework

The goal of data-centric approaches is to augment training datasets by generating useful labeled data examples. Under that, the goal of our data-centric transfer (DCT) framework is to *transfer* the knowledge from unlabeled data into the data augmentation. Specifically, for each graph-label pair $(G^{[k]} \in \mathcal{G}^{[k]}, y^{[k]} \in \mathcal{Y}^{[k]})$ in the task $k$, the framework is expected to output a new example $G'^{[k]}$ with the label $y'^{[k]}$ such that (1) $y'^{[k]} = y^{[k]}$ and (2) $G'^{[k]}$ and $G^{[k]}$ are from the same graph space $\mathcal{G}^{[k]}$. However, if the graph structures of $G'^{[k]}$ and $G^{[k]}$ were too similar, the augmentation would duplicate the original data examples, become useless, and even cause over-fitting. So, the optimal graph data augmentation should *enrich the training data with good diversity as well as preserve the labels of the original graphs*. To achieve this, DCT utilizes a diffusion probabilistic model to first *learn the data distribution from unlabeled graphs* (Section 3.2). Then DCT adapts the reverse process in the diffusion model to *generate task-specific labeled graphs for data augmentation* (Section 3.3). Thus, the augmented graphs will be derived from the distribution of a huge collection of unlabeled data for *diversity*. To *preserve the labels*, DCT controls the reverse process with two task-related optimization objectives to transfer *minimal sufficient knowledge* from the unlabeled data. The first objective minimizes an upper bound of mutual information between the augmented and the original graphs in the graph space. The second objective maximizes the probability of the predicted label of augmented graphs being the same as the label of original graphs. The first is for minimal knowledge transfer, and the second is for sufficient knowledge transfer. DCT integrates the two objectives into the reverse process of the diffusion model to guide the generation of new labeled graphs. DCT iteratively trains the graph property predictor (used in the second objective) and creates the augmented training data. To simplify notations, we remove the task superscript $[k]$ in the following sections.

### 3.2 Learning Data Distribution from Unlabeled Graphs

The diffusion process for graphs in Figure 2 applies to both graph structure and node features. The diffusion model slowly corrupts unlabeled graphs to a standard normal distribution with noise. For

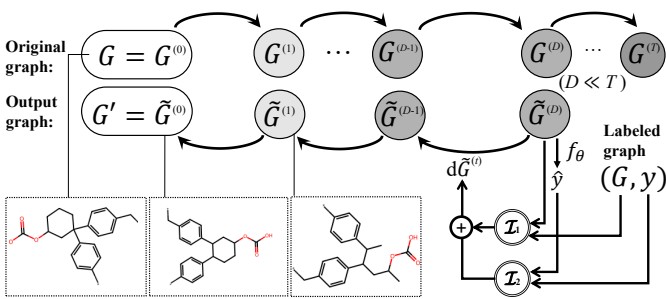

Figure 2: Diffusion model in DCT: It performs task-specific data augmentation using objectives $\mathcal{I}_1$ and $\mathcal{I}_2$ in the reverse process. The model was trained on unlabeled graphs to learn the general data distribution. Then it generates $(G', y' = y)$ based on $(G, y)$ in the reverse process. It perturbs $G$ with $D$ steps and optimizes $G'$ to be minimally similar to $G$ (Objective $\mathcal{I}_1$) and sufficiently preserve the label of $G$ (Objective $\mathcal{I}_2$).

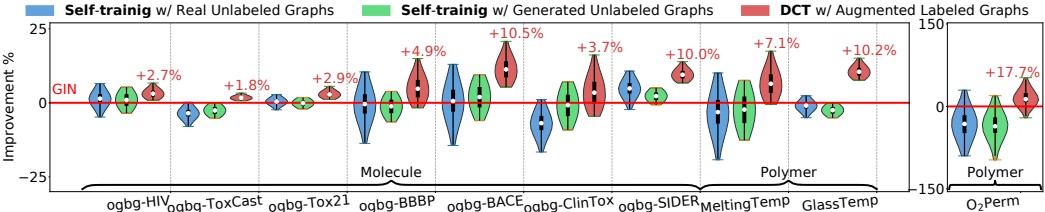

Figure 3: Relative improvement (increased AUC or reduced MAE) from three data-centric methods (over ten runs), compared to the basic GIN: Blue is for self-training with selected real unlabeled graphs. Green is for self-training with graphs directly generated by a standard diffusion model. Red is for DCT that generates task-specific labeled graphs. The first two often make little or negative impact. Our DCT has consistent and significant improvement shown as the percentages in red.

graph generation, the model samples noise from the normal distribution and learns a score function to reverse the perturbed noise. Given an unlabeled graph $G$, we use continuous time $t \in [0, T]$ to index multiple diffusion steps $\{G^{(t)}\}_{t=1}^{T}$ on the graph, such that $G^{(0)}$ follows the original data distribution and $G^{(T)}$ follows a prior distribution like the normal distribution. The forward diffusion is a stochastic differential equation (SDE) from the graph to the noise:

$$dG^{(t)} = \mathbf{f}\left(G^{(t)}, t\right) dt + g(t) \, d\mathbf{w}, \tag{3}$$

where $\mathbf{w}$ is the standard Wiener process, $\mathbf{f}(\cdot, t) : \mathcal{G} \to \mathcal{G}$ is the drift coefficient and $g(t) : \mathbb{R} \to \mathbb{R}$ is the diffusion coefficient. $\mathbf{f}(G^{(t)}, t)$ and $g(t)$ relate to the amount of noise added to the graph at each infinitesimal step $t$. The reverse-time SDE uses gradient fields or scores of the perturbed graphs $\nabla_{G^{(t)}} \log p_t(G^{(t)})$ for denoising and graph generation from $T$ to $0$ (Song et al., 2021):

$$dG^{(t)} = \left[\mathbf{f}(G^{(t)}, t) - g(t)^2 \nabla_{G^{(t)}} \log p_t(G^{(t)})\right] dt + g(t) d\overline{\mathbf{w}}, \tag{4}$$

where $p_t(G^{(t)})$ is the marginal distribution at time $t$ in forward diffusion. $\overline{\mathbf{w}}$ is a reverse time standard Wiener process. $dt$ here is an infinitesimal negative time step. The score $\nabla_{G^{(t)}} \log p_t(G^{(t)})$ is unknown in practice and it is approximated by the score function $\mathbf{s}(G^{(t)}, t)$ with score matching techniques (Song et al., 2021). On graphs, Jo et al. (2022) used two GNNs to develop the score function $\mathbf{s}(G^{(t)}, t)$ to de-noise both node features and graph structures and details are in appendix B.3.

### 3.3 Generating Task-specific Labeled Graphs

*Self-training* approaches would propose to either (1) select unlabeled graphs by a graph property predictor or (2) generate graphs directly from the standard diffusion model, and then use the predictor to assign them labels so that the training data could be enriched. However, we have observed that neither of them can guarantee positive impact on the prediction performance. In fact, as shown in Figure 3, they make very little or even negative impact. That is because *the selected or directly-generated graphs are too different from the labeled graph space of the target tasks*. Task details of ten datasets are in appendix C.

Given a labeled graph $(G, y)$ from the original dataset of a specific task, the new labeled graph $(G', y')$ is expected to provide *useful knowledge to augment* the training set. We name it *the augmented graph* throughout this section. The augmented graph is desired to have the following two properties, as in Section 3.1: **Task relatedness**: As an effective training data point, $G' \in \mathcal{G}$ and $y' \in \mathcal{Y}$ are from the graph/label spaces of the specific task where $(G, y)$ come from and thus transfer sufficient task knowledge into the training set; **Diversity**: If $G'$ was too similar to $G$, the new data point would cause severe over-fitting on the property prediction model. The augmentation aims to learn from unlabeled graph to create diverse data points, which should contain minimal task knowledge about $G$.

The selected unlabeled graphs used in *self-training* have little task relatedness because the unlabeled data distribution might be too far from the one of the specific task. Existing graph *data augmentation* methods could not create diverse graph examples because they manipulated labeled graphs and did not learn from the unlabeled graphs. Our novel data-centric approach DCT works towards both desired properties by transferring *minimally sufficient knowledge* from the unlabeled graphs: **Sufficiency** is achieved by maximizing the possibility for label preservation (i.e., $y' = y$). It ensures that the knowledge from unlabeled graphs is task-related; **Minimality** refers to the minimization of graph similarity between $G'$ and $G$ to ensure that the augmentation introduces diversity. Both optimizations can be formulated using mutual information $\mathcal{I}(\cdot\,;\cdot)$ to generate task-specific labeled data $(G', y')$:

**Definition 3.1** (Sufficiency for Data Augmentation). The augmented graph $G'$ sufficiently preserves the label of the original graph $G$ if and only if $\mathcal{I}(G'; y) = \mathcal{I}(G; y)$.

**Definition 3.2** (Minimal Sufficiency for Data Augmentation). The Sufficiency is minimal for data augmentation if and only if $\mathcal{I}(G'; G) \leq \mathcal{I}(\bar{G}; G), \forall \bar{G}$ represents any augmented graph that sufficiently preserves the original graph's label.

Self-supervised tasks applied a similar philosophy in pre-training (Soatto and Chiuso, 2016), however, they did not use labeled data from any specific tasks. So the optimizations were unable to extract useful knowledge and transfer it to the downstream (Tian et al., 2020). In our DCT that performs task-specific data augmentation, the augmented graphs can be optimized toward the objectives using any labeled graph $G$ and its label $y$:

$$\min_{\mathcal{I}_1} \max_{\mathcal{I}_2} \; \mathbb{E}_G \left[ \mathcal{I}_1 \left( G'; G \right) + \mathcal{I}_2 \left( G'; y \right) \right]. \tag{5}$$

For the first objective, we use the leave-one-out variant of InfoNCE (Poole et al., 2019; Oord et al., 2018) as the upper bound estimation. For the $i$-th labeled graph $(G_i, y_i)$,

$$\mathcal{I}_1 \leq \mathcal{I}_{\text{bound}}(G'_i; G_i) = \log \frac{p(G'_i|G_i)}{\sum_{j=1, j\neq i}^{M} p(G'_i|G_j)}, \tag{6}$$

where $G'_i$ is the augmented graph. When $G'_i$ is optimized, $G_i$ makes a positive pair; $\{G_j\}$ ($j \neq i$) are $M-1$ negative samples of labels that do not equal $y_i$. ($M$ is a hyperparameter.) We use cosine similarity and a softmax function to calculate $p(G'_i|G_j) = \frac{\exp(\text{sim}(G'_i, G_j))}{\sum_{j=1}^{M} \exp(\text{sim}(G'_i, G_j))}$. In practice, we extract statistical features of graphs to calculate their similarity. Details are in appendix B.2.

For the second objective, we denote the predicted label of the augmented graph $G'$ by $f_\theta(G')$. We maximize the log likelihood $\log p(y|f_\theta(G'))$ to maximize $\mathcal{I}_2(G'; y)$. Specifically, after the predictor $f_\theta$ is trained for several epochs on the labeled data, we freeze its parameters and use it to optimize the augmented graphs so they are task-related:

$$\mathcal{L}(G') = \mathcal{I}_{\text{bound}} \left( G'; G \right) - \log p \left( y|f_\theta(G') \right). \tag{7}$$

**Framework details:** As shown in Figure 2, after the diffusion model learns the data distribution from unlabeled graphs, given a labeled graph $G$ from a specific task, DCT perturbs it for $D$ ($D \ll T$) steps. The perturbed noisy graph, denoted by $\tilde{G}^{(D)}$, stays inside the task-specific graph and label space, rather than the noise distribution (at step $T$). To reverse the noise in it and generate a task-specific augmented example $G'$, DCT integrates the loss function in Eq. (7) into the score function $\mathbf{s}(\cdot, t)$ for minimal sufficient knowledge transfer:

$$\mathrm{d}\tilde{G}^{(t)} = \left[ \mathbf{f}(\tilde{G}^{(t)}, t) - g(t)^2 \left( \mathbf{s}(\tilde{G}^{(t)}, t) - \alpha \nabla_{\tilde{G}^{(t)}} \mathcal{L}(\tilde{G}^{(t)}) \right) \right] \mathrm{d}t + g(t)\mathrm{d}\overline{\mathbf{w}}, \tag{8}$$

where $\alpha$ is a scalar for score alignment between $\mathbf{s}$ and $\nabla \mathcal{L}$ to avoid the dominance of any of them: $\alpha = \frac{\|\mathbf{s}(\tilde{G}^{(t)}, t)\|_2}{\|\nabla_{\tilde{G}^{(t)}} \mathcal{L}(\tilde{G}^{(t)})\|_2}$. Because $\tilde{G}^{(t)}$ is an intermediate state in the reverse process, the noise in it

Table 1: Statistics of datasets for graph property prediction in different domains.

| Data Type | Dataset | # Graphs | Prediction Task | # Task | Avg./Max # Nodes | Avg./Max # Edges |
|-----------|---------|----------|-----------------|--------|------------------|------------------|
| Molecules | ogbg-HIV | 41,127 | Classification | 1 | 25.5 / 222 | 54.9 / 502 |
|  | ogbg-ToxCast | 8,576 | Classification | 617 | 18.8 / 124 | 38.5 / 268 |
|  | ogbg-Tox21 | 7,831 | Classification | 12 | 18.6 / 132 | 38.6 / 290 |
|  | ogbg-BBBP | 2,039 | Classification | 1 | 24.1 / 132 | 51.9 / 290 |
|  | ogbg-BACE | 1,513 | Classification | 1 | 34.1 / 97 | 73.7 / 202 |
|  | ogbg-ClinTox | 1,477 | Classification | 2 | 26.2 / 136 | 55.8 / 286 |
|  | ogbg-SIDER | 1,427 | Classification | 27 | 33.6 / 492 | 70.7 / 1010 |
|  | ogbg-Lipo | 4200 | Regression | 1 | 27 / 115 | 59 / 236 |
|  | ogbg-ESOL | 1128 | Regression | 1 | 13.3 / 55 | 27.4 / 124 |
|  | ogbg-FreeSolv | 642 | Regression | 1 | 8.7 / 24 | 16.8 / 50 |
| Polymers | GlassTemp | 7,174 | Regression | 1 | 36.7 / 166 | 79.3 / 362 |
|  | MeltingTemp | 3,651 | Regression | 1 | 26.9 / 102 | 55.4 / 212 |
|  | ThermCond | 759 | Regression | 1 | 21.3 / 71 | 42.3 / 162 |
|  | $O_2$Perm | 595 | Regression | 1 | 37.3 / 103 | 82.1 / 234 |
| Proteins | PPI | 88000 | Classification | 40 | 49.4 / 111 | 890.8 / 11556 |

may fail the optimizations. So, we design a new sampling method named *double-loop sampling* for accurate loss calculation. It has an inner-loop sampling using Eq. (4) to sample $\hat{G}_{(t)}$, as the denoised version of $\tilde{G}^{(t)}$ at the reverse time $t$. Then $\nabla_{\hat{G}}\mathcal{L}(\hat{G}_{(t)})$ is calculated as an alternative for $\nabla_{\tilde{G}^{(t)}}\mathcal{L}(\tilde{G}^{(t)})$. Finally, an outer-loop sampling takes one step to guide denoising using Eq. (8).

DCT iteratively creates the augmented graphs $(G', y')$, updates the training dataset $\{(G_i, y_i)\}$, and trains the graph property predictor $f_\theta$. In each iteration, for task $k$, $n \ll N^{[k]}$ labeled graphs of the lowest property prediction loss are selected to create the augmented graphs. The predictor is better fitted to these graphs for more accurate sufficiency estimation of the augmentation.

## 4 Experiments

In this section, we present and analyze experimental results to demonstrate the outstanding performance of DCT, the usefulness of new optimization objectives, the effect of hyperparameters and iterative process, and the interpretability of "visible" knowledge transfer from unlabeled graphs.

### 4.1 Experimental Setup

**Tasks and metrics:** Experiments are conducted on 15 graph property prediction tasks in chemistry, material science, and biology, including seven molecule classification, three molecule regression tasks from open graph benchmarks (Hu et al., 2020), four polymer regression tasks, and protein function prediction (PPI) (Hu et al., 2019). Dataset statistics is presented in Table 1. We use the area under the ROC curve (AUC) to evaluate classifiers and mean absolute error (MAE) for regressors.

**Baselines and implementation:** Besides GIN, there are three lines of baseline methods: (1) *self-supervised learning methods* including EDGEPRED, ATTRMASK, CONTEXTPRED in (Hu et al., 2019), INFOMAX (Velickovic et al., 2019), JOAO (You et al., 2021), GRAPHLOG (Xu et al., 2021), MGSSL Zhang et al. (2021) and D-SLA (Kim et al., 2022), (2) *semi-supervised learning methods* including self-training with selected unlabeled graphs (ST-REAL) and generated graphs (ST-GEN) and INFOGRAPH (Sun et al., 2020), and (3) *graph data augmentation (GDA) methods* including FLAG (Kong et al., 2022), GREA (Liu et al., 2022a), and G-MIXUP (Han et al., 2022). For self-supervised pre-training, we follow their own settings and directly use their pre-trained models if available. For semi-supervised learning methods and DCT, we use 113K QM9 (Ramakrishnan et al., 2014) and 306K PPI graphs (Hu et al., 2019) as unlabeled data sources for the tasks on molecules/polymers and proteins, respectively. For DCT, we tune three major hyper-parameters: the number of perturbation steps $D \in [1, 10]$, the number of negative samples $M \in [1, 10]$, and top-$n$ % labeled graphs of lowest property prediction loss selected for data augmentation.

### 4.2 Outstanding Property Prediction Performance

We report the model performance using mean and standard deviation over 10 runs Table 2. DCT is the best on all 15 tasks compared to the state-of-the-art baselines. Our observations are:

Table 2: Mean(Std) on tasks from different fields. The best mean is **bold**. The best baseline is underlined. Results are highlighted if unlabeled graphs bring significant negative impacts compared to GIN. The MAE for ThermCond is scaled × 100. G-MixUp was proposed for classification. MGSSL was proposed for molecules.

| | | Molecule Classification: AUC (%) ↑ | | | | | | |
| | # Training Graphs | ogbg-HIV 32,901 | ogbg-ToxCast 6,860 | ogbg-Tox21 6,264 | ogbg-BBBP 1,631 | ogbg-BACE 1,210 | ogbg-ClinTox 1,181 | ogbg-SIDER 1,141 |
|---|---|---|---|---|---|---|---|---|
| | GIN | 77.4(1.2) | 66.9(0.2) | 76.0(0.6) | 67.5(2.7) | 77.5(2.8) | 88.8(3.8) | 58.1(0.9) |
| Self-Supervised | EdgePred | 78.1(1.3) | 63.9(0.4) | 75.5(0.4) | 69.9(0.5) | 79.5(1.0) | 62.9(2.3) | 59.7(0.8) |
| | AttrMask | 77.1(1.7) | 64.2(0.5) | 76.6(0.4) | 63.9(1.2) | 79.3(0.7) | 70.4(1.1) | 60.7(0.4) |
| | ContextPred | 78.4(0.1) | 63.7(0.3) | 75.0(0.1) | 68.8(1.6) | 75.7(1.0) | 63.2(6.5) | 60.7(0.8) |
| | InfoMax | 75.4(1.8) | 61.7(1.0) | 75.5(0.4) | 69.2(0.5) | 76.8(0.2) | 73.0(0.2) | 58.6(0.5) |
| | JOAO | 76.2(0.2) | 64.8(0.3) | 74.8(0.5) | 69.3(2.5) | 75.9(3.9) | 69.4(4.5) | 60.8(0.6) |
| | GraphLog | 74.8(1.1) | 63.2(0.8) | 75.4(0.8) | 67.5(2.3) | 80.4(3.6) | 69.0(6.6) | 57.0(0.9) |
| | MGSSL | 77.1(1.1) | 65.7(0.4) | 77.2(0.3) | 66.9(0.9) | 81.3(2.4) | 69.8(5.0) | 63.6(1.0) |
| | D-SLA | 76.9(0.9) | 60.8(1.2) | 76.1(0.1) | 62.6(1.0) | 80.3(0.6) | 78.3(2.4) | 55.1(1.0) |
| Semi-SL | InfoGraph | 73.3(0.7) | 61.5(1.1) | 67.6(0.9) | 61.6(4.4) | 75.9(1.8) | 62.2(5.5) | 56.3(2.3) |
| | ST-Real | 78.3(0.6) | 64.5(1.0) | 76.2(0.5) | 66.7(1.9) | 77.4(1.8) | 82.2(2.4) | 60.8(1.2) |
| | ST-Gen | 77.9(1.6) | 65.1(1.0) | 75.8(0.9) | 66.3(1.5) | 78.4(3.0) | 87.3(1.3) | 59.3(1.3) |
| GDA | FLAG | 74.6(1.7) | 59.9(1.6) | 76.9(0.7) | 66.6(1.0) | 79.1(1.2) | 85.1(3.4) | 57.6(2.3) |
| | GREA | 79.3(0.9) | 67.5(0.7) | 77.2(1.2) | 69.7(1.3) | 82.4(2.4) | 87.9(3.7) | 60.1(2.0) |
| | G-MixUp | 77.1(1.1) | 55.6(1.1) | 64.6(0.4) | 70.2(1.0) | 77.8(3.3) | 60.2(7.5) | 56.8(3.5) |
| | DCT (Ours) | **79.5**(1.0) | **68.1**(0.2) | **78.2**(0.2) | **70.8**(0.5) | **85.6**(0.6) | **92.1**(0.8) | **63.9**(0.3) |

| | | Molecule Regression: MAE ↓ | | | Polymer Regression: MAE ↓ | | | | Bio: AUC (%)↑ |
| | # Training Graphs | ogbg-Lipo 3,360 | ogbg-ESOL 902 | ogbg-FreeSolv 513 | GlassTemp 4,303 | MeltingTemp 2,189 | ThermCond 455 | $O_2$Perm 356 | PPI 60,715 |
|---|---|---|---|---|---|---|---|---|---|
| | GIN | 0.545(0.019) | 0.766(0.016) | 1.639(0.146) | 26.4(0.2) | 40.9(2.2) | 3.25(0.19) | 201.3(45.0) | 69.1(0.0) |
| Self-Supervised | EdgePred | 0.585(0.008) | 1.062(0.066) | 2.249(0.150) | 27.6(1.4) | 47.4(2.8) | 3.69(0.50) | 207.3(41.7) | 63.7(1.1) |
| | AttrMask | 0.573(0.009) | 1.041(0.041) | 1.952(0.088) | 27.7(0.8) | 45.8(2.6) | 3.17(0.32) | 179.9(30.8) | 64.1(1.8) |
| | ContextPred | 0.592(0.007) | 0.971(0.027) | 2.193(0.151) | 27.6(0.3) | 46.7(1.9) | 3.15(0.24) | 191.2(35.2) | 62.0(1.2) |
| | InfoMax | 0.581(0.009) | 0.935(0.018) | 2.197(0.129) | 27.5(0.8) | 46.5(2.8) | 3.31(0.25) | 231.0(52.6) | 63.3(1.2) |
| | JOAO | 0.596(0.016) | 1.098(0.037) | 2.465(0.095) | 27.5(0.2) | 46.0(0.2) | 3.55(0.26) | 207.7(43.7) | 61.5(1.2) |
| | GraphLog | 0.577(0.010) | 1.109(0.059) | 2.373(0.283) | 29.5(1.3) | 50.3(3.3) | 3.01(0.17) | 229.7(48.3) | 62.1(0.6) |
| | MGSSL | 0.569(0.007) | 0.998(0.031) | 1.956(0.077) | 26.9(0.4) | 42.7(1.2) | 3.10(0.14) | 201.1(31.9) | N.A. |
| | D-SLA | 0.563(0.004) | 1.064(0.030) | 2.190(0.149) | 27.5(1.0) | 51.7(2.5) | 2.71(0.08) | 257.8(30.2) | 65.0(1.2) |
| Semi-SL | InfoGraph | 0.793(0.094) | 1.285(0.093) | 3.710(0.418) | 30.8(1.2) | 51.2(5.1) | 2.75(0.15) | 207.2(21.8) | 67.7(0.4) |
| | ST-Real | 0.526(0.009) | 0.788(0.070) | 1.770(0.251) | 26.6(0.3) | 42.3(1.2) | 2.64(0.07) | 256.0(17.5) | 68.9(0.1) |
| | ST-Gen | 0.531(0.031) | 0.724(0.082) | 1.547(0.082) | 26.8(0.3) | 42.0(0.9) | 2.70(0.03) | 262.2(10.1) | 68.6(0.6) |
| GDA | FLAG | 0.528(0.012) | 0.755(0.039) | 1.565(0.098) | 26.6(1.3) | 44.2(2.0) | 3.05(0.10) | 177.7(60.7) | 69.2(0.2) |
| | GREA | 0.586(0.036) | 0.805(0.135) | 1.829(0.368) | 26.7(1.0) | 41.1(0.8) | 3.23(0.18) | 194.0(45.5) | 68.8(0.2) |
| | DCT (Ours) | **0.516**(0.071) | **0.717**(0.020) | **1.339**(0.075) | **23.7**(0.2) | **38.0**(0.8) | **2.59**(0.11) | **165.6**(24.3) | **69.5**(0.2) |

**(1) GIN is the most competitive baseline and outperforms self-supervised learning methods.** On 7 of 15 tasks, GIN outperforms all the 7 self-supervised learning methods. Because self-supervised pre-training imposes constraints on the model architecture, it undermines the true power of GNNs and under-performs the GNNs that are properly used.

**(2) Self-training and GDA methods perform better than GIN but cannot effectively learn from unlabeled data.** Self-training (ST-Real and ST-Gen) is often the best baseline in regression tasks. GDA (GREA and G-MixUp) methods outperform self-training in most classification tasks except ogbg-SIDER, because they are often designed to exploit categorical labeled data and remain under-explored for regression. Although self-training benefits from selecting unlabeled examples in some graph regression tasks, they are *negatively* affected by the unlabeled graphs in the classification tasks such as ogbg-ToxCast and ogbg-ClinTox. As indicated in Figure 3, it is inappropriate to pseudo-label unlabeled graphs in self-training due to the huge gap between the unlabeled data and target task.

**(3) DCT transfers useful knowledge from unlabeled data by data augmentation.** DCT outperforms the best baseline relatively by +3.9%, +13.4%, and +10.2% when there are only 1,210, 513, and 4,303 training graphs on ogbg-BACE, ogbg-FreeSolv, and GlassTemp, respectively. Compared to the self-supervised baselines, the improvement from DCT is more significant, so the knowledge transfer is more effective. For example, on ogbg-FreeSolv and $O_2$Perm, DCT performs better than the best self-supervised baselines relatively by +45.8% and +8.0%, respectively. On regression tasks that involve knowledge transfer across domains (*e.g.,* from molecules to polymers), DCT reduces MAE relatively by 1.9% ∼ 10.2% compared to the best baseline. All these results demonstrate the outstanding performance of task-specific data augmentation in DCT.

Table 3: Comprehensive ablation studies for DCT on tasks ogbg-BACE, ogbg-SIDER, ogbg-FreeSolv, and $O_2$Perm. Objectives include minimizing $\mathcal{I}_1(G', G)$ and/or maximizing $\mathcal{I}_2(G', y)$.

| | | Objectives | | Classification | | Regression | |
|---|---|---|---|---|---|---|---|
| | | $\mathcal{I}_1(G',G)$ | $\mathcal{I}_2(G',y)$ | BACE | SIDER | FreeSolv | $O_2$Perm |
| | | Top Baseline Method | | $82.4_{(2.4)}$ | $60.8_{(1.2)}$ | $1.547_{(0.082)}$ | $177.7_{(60.7)}$ |
| Unlabeled Data Sources | QM9 | ✗ | ✗ | $84.4_{(2.6)}$ | $63.7_{(0.3)}$ | $1.473_{(0.192)}$ | $177.4_{(27.3)}$ |
| | | ✓ | ✗ | $85.2_{(1.3)}$ | $63.7_{(0.2)}$ | $1.415_{(0.145)}$ | $171.4_{(14.0)}$ |
| | | ✗ | ✓ | $84.7_{(1.8)}$ | $63.8_{(0.5)}$ | $1.344_{(0.096)}$ | $172.6_{(32.9)}$ |
| | | ✓ | ✓ | $\mathbf{85.6}_{(0.6)}$ | $\mathbf{63.9}_{(0.3)}$ | $\mathbf{1.339}_{(0.075)}$ | $\mathbf{165.6}_{(24.3)}$ |
| | ZINC | ✗ | ✗ | $82.8_{(1.8)}$ | $63.5_{(0.7)}$ | $1.524_{(0.219)}$ | $175.5_{(11.9)}$ |
| | | ✓ | ✗ | $83.3_{(2.2)}$ | $63.5_{(0.7)}$ | $1.455_{(0.207)}$ | $172.4_{(60.8)}$ |
| | | ✗ | ✓ | $84.3_{(0.6)}$ | $63.5_{(0.6)}$ | $1.514_{(0.214)}$ | $171.5_{(26.0)}$ |
| | | ✓ | ✓ | $84.9_{(0.4)}$ | $63.7_{(0.7)}$ | $1.408_{(0.092)}$ | $169.3_{(15.3)}$ |

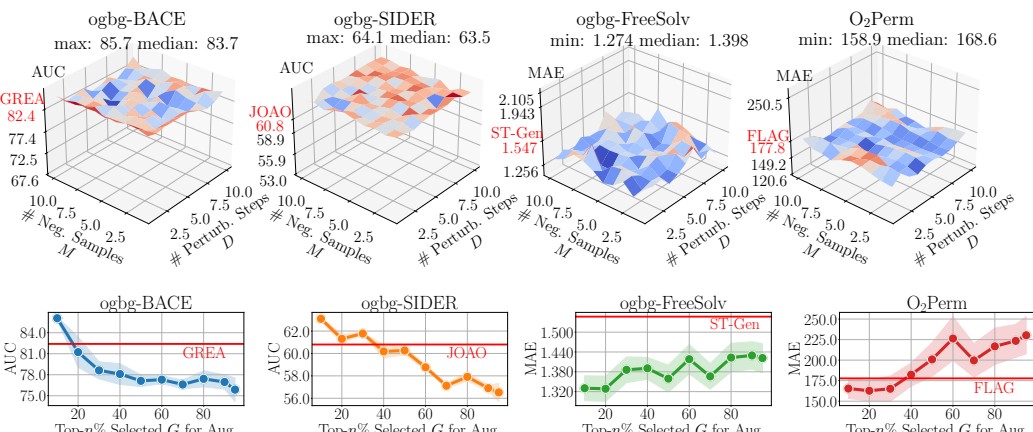

Figure 4: Effect of hyper-parameters, including the number of perturbation steps $D \in [1, 10]$, the number of negative graphs $M \in [1, 10]$, and top-$n$ % labeled graphs whose labels are predicted the most accurately and that are selected for data augmentation, where $n \in [10, 100]$.

## 4.3 Ablation Studies and Performance Analysis

**Comprehensive ablation studies:** In Table 3, we investigate how the task-related objectives in Eq. (5) impact the performance of DCT. First, DCT outperforms the top baseline even if the two task-related optimization objectives are disabled. This is because DCT generates new training examples based on original labeled graphs: the data augmentation has already improved the diversity of the training dataset a little bit. Second, adding the objective $\mathcal{I}_1$ further improves the performance by encouraging the generation of diverse examples, because it minimizes the similarity between the original graph and augmented graph in the graph space. Third, we receive the best performance of DCT when it combines $\mathcal{I}_1$ and $\mathcal{I}_2$ objectives to generate task-related and diverse augmented graphs. When we change the unlabeled data source from QM9 to the ZINC dataset from (Jo et al., 2022), similar observations confirm the necessity of the task-related objectives.

**Effect of hyper-parameters:** The impacts of three hyper-parameters of DCT are studied: the number of perturbation steps $D$, the number of negative samples $M$ in Eq. (6), and the number of augmented graphs in each iteration (*i.e.,* top-$n$ % selected graph for augmentation). Results from Figure 4 show that DCT is robust to a wide range of $D$ and $M$ valued from 0 to 10. They suggest that $D$ and $M$ can be set as 5 in most cases. As for the number of the augmented graphs in each iteration, results show that noisy graphs are often created when $n$ is higher than 30%, because the predictor cannot effectively guide the data augmentation for those labeled graphs whose labels are hard to predict. So, 10% is suggested as the default of top-$n$%.

**Iterative process:** Figure 5 investigates the relationship between the quality of augmented graphs and the accuracy of property prediction models. We save a predictor checkpoint every 20 epochs

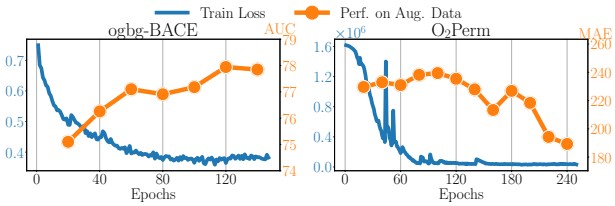

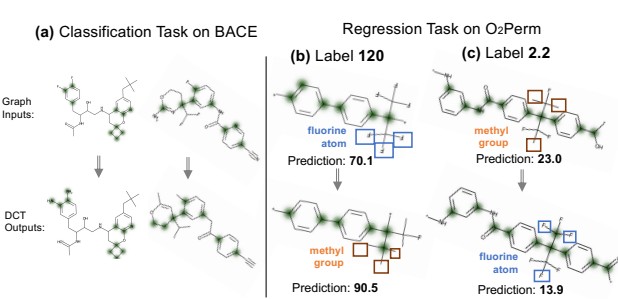

Figure 5: Data augmentation and model training mutually enhance each other over epochs. The predictor is saved every 20 epochs to guide the generation of augmented graphs. The performance of GIN trained on these augmented graphs reflects the quality of the augmented data.

Figure 6: Case studies of augmented graphs. The green highlighted subgraphs are from GIN with top-k pooling. Examples show that the augmented graphs from DCT preserve the core structures of original graphs. Key concepts in the unlabeled graphs like chemical validity are transferred to downstream tasks. Domain knowledge such as the relationship between the permeability and the fluorine atom/methyl group is captured to guide task-specific generation.

to guide the generation of the augmented examples. We evaluate the quality of augmented graphs by using them to train GIN and report AUC/MAE. The data augmentation gradually decreases the training loss of property prediction. On the other hand, the increased GIN performance indicates that the quality of augmented examples is also improved over epochs. The data augmentation and predictor training mutual enhance each other.

## 4.4 Interpretability of Visible Knowledge Transfer

Knowledge transfer by data augmentation gives visible examples, allowing us to study what is learned. We visualize a few augmented graphs in DCT using ogbg-BACE and $O_2$Perm. We adapt top-k pooling (Knyazev et al., 2019) to select the subgraphs that GIN used for prediction. The selected subgraphs are highlighted in green in Figure 6. The three examples show that *the augmented graphs can identify and preserve the core structures* that GIN uses to predict property values. These augmented graphs are chemically valid, showing that *concepts such as some chemical rules from the unlabeled graphs are successfully transferred to downstream tasks*. More results are in appendix D.2. Regarding task-specific knowledge, it is known that the fluorine atom and the methyl group are usually negatively and positively correlated to the permeability, respectively (Park et al., 2003; Corrado and Guo, 2020). The augmented examples show that DCT *captures this domain knowledge with the task-related objectives*. In example (b), DCT replaces most of the fluorine atoms with the methyl groups. It encourages GIN to learn the positive relationship between the methyl group and the permeability so that GIN predicts a high label value. In example (c), DCT replaces the methyl groups with fluorine atoms. It encourages GIN to learn the negative relationship between the fluorine atom and the permeability so that GIN predicts a low label value.

## 5 Related Work

### 5.1 Graph Property Prediction

Graph neural networks (GNNs) (Kipf and Welling, 2017; Xu et al., 2019) are commonly used for graph property prediction in chemistry and polymer informatics tasks (Otsuka et al., 2011; Hu et al., 2020; Zhou et al., 2022). However, it is hard to annotate enough labels in these domains. Recent work used *self-supervised tasks* such as node attribute prediction and graph structure prediction (Hu et al., 2019; You et al., 2021; Kim et al., 2022) to pre-train architecture-fixed GNNs. Sun et al. (2022) observed that the existing methods might fail to transfer knowledge from unlabeled graph data. Flexible GNN architectures for downstream tasks would be desirable.

*Graph data augmentation* (GDA) methods do not restrict GNN architecture choices to improve prediction accuracy (Trivedi et al., 2022; Zhao et al., 2022, 2023; Ding et al., 2022). They learn to create new examples that preserve the properties of original graphs (Liu et al., 2022b,a; Kong et al., 2022; Han et al., 2022; Luo et al., 2022). However, they purely manipulate labeled examples and thus *cannot utilize the knowledge in unlabeled graphs*. Our DCT combines the knowledge from the unlabeled dataset and the labeled task dataset. It creates label-preserved graph examples with the knowledge transferred from the unlabeled data. It allows the GNN models to have flexible architectures.

## 5.2 Learning from Unlabeled Data

*Pre-training on self-supervised tasks* such as masked image modeling and autoregressive text generation is effective for large language and vision models (Brown et al., 2020; He et al., 2022). However, the hand-crafted self-supervised tasks could hardly help models learn useful knowledge from unlabeled graphs *due to the gap between these label-agnostic tasks and the downstream prediction tasks* towards drug discovery and material discovery (Sun et al., 2021; Kim et al., 2022; Inae et al., 2023). A universal self-supervised task to learn from the unlabeled graphs remains under-explored (Sun et al., 2022; Trivedi et al., 2022).

*Semi-supervised learning* assumes that unlabeled and labeled data are from the same source (Liu et al., 2023). The learning objective in the latent space is usually mutual information maximization that encourages similarity between the representations of unlabeled and labeled graphs (Sun et al., 2020). However, *the distributions of the unlabeled and labeled data could be very different* due to the different types of sources (Hu et al., 2019), leading to negative impacts on the property prediction on the labeled graphs. *Self-training*, as a specific type of semi-supervised learning method, selects the unlabeled graphs of confidently predictable labels and assigns pseudo-labels for them (Lee et al., 2013; Iscen et al., 2019). Many studies have explored improving uncertainty estimation (Gal and Ghahramani, 2016; Tagasovska and Lopez-Paz, 2019; Amini et al., 2020) to help the model filter out noise for reliable pseudo-labels. Recently, pseudo-labels have been applied in imbalanced learning (Liu et al., 2023) and representation learning (Ghiasi et al., 2021). However, self-training is restricted to confidently predictable labels and may ignore the huge number of any other unlabeled graphs (Huang et al., 2022). Therefore, it cannot fully utilize the knowledge in the unlabeled graphs.

In contrast, our DCT employs a diffusion model to extract knowledge (as the diffusion and reverse processes) from *all the unlabeled graphs*. DCT represents the knowledge as task-specific labeled examples to augment the target dataset, instead of uninterpretable pre-trained model parameters. We note that self- or semi-supervised learning does not conflict with DCT, and we leave their combinations for future work.

## 5.3 Diffusion Models on Graphs

Recent works have improved the diffusion models on graphs (Niu et al., 2020; Jo et al., 2022; Vignac et al., 2022; Kong et al., 2023; Chen et al., 2023). EDP-GNN (Niu et al., 2020) employed score matching for permutation-invariant graph data distribution. GDSS (Jo et al., 2022) extended the continuous-time framework [6] to model node-edge joint distribution. DiGress (Vignac et al., 2022) used the transition matrix to preserve the discrete natures of the graph structure. GraphARM (Kong et al., 2023) introduced a node-absorbing autoregressive diffusion process. EDGE (Chen et al., 2023) focused on efficiently generating larger graphs. Instead of improving the generation performance of the diffusion model, our model builds on score-based diffusion models (Jo et al., 2022; Song et al., 2021) for predictive tasks, *i.e.,* , graph classification and graph regression.

## 6 Conclusion

In this work, we made the first attempt to transfer minimal sufficient knowledge from unlabeled graphs by data augmentation. We proposed a data-centric framework to use the diffusion model trained on the unlabeled graphs and use two task-related objectives to generate task-specific augmented graphs. Experiments demonstrated the performance of the proposed framework through visible augmented examples. It is better than self-supervised learning, self-training, and graph data augmentation methods on as many as 15 tasks.

## Acknowledgments and Disclosure of Funding

This work was supported by NSF IIS-2142827, IIS-2146761, IIS-2234058, CBET-2102592, and ONR N00014-22-1-2507.

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

## A    Additional Related Work on Data-Centric Approach

**Data Augmentation**    Data augmentation creates new examples with preserved labels but uses no unlabeled data (Shorten and Khoshgoftaar, 2019; Kashefi and Hwa, 2020; Balestriero et al., 2022). Examples of heuristic data augmentation techniques include flipping, distorting, and rotating images (Shorten and Khoshgoftaar, 2019), using lexical substitution, inserting words, and shuffling sentences in texts (Kashefi and Hwa, 2020), and deleting nodes and dropping edges in graphs (Zhao et al., 2021, 2023). While human knowledge can be used to improve data diversity and reduce over-fitting in heuristic methods, it is difficult to use a single heuristic method to preserve the different labels for different tasks (Balestriero et al., 2022; Cubuk et al., 2019). So, automated augmentation (Cubuk et al., 2019) learned from data to search for the best policy to combine a bunch of predefined heuristic augmentations. Generation models (Antoniou et al., 2017; Bowles et al., 2018; Han et al., 2022) create in-class examples. Other learning ideas such as FATTEN (Liu et al., 2018) and GREA (Liu et al., 2022a) learned to split the latent space for data augmentation. However, learning and augmenting from insufficient labels at the same time may limit the diversity of new examples and cause over-fitting. DCT leverages unlabeled data to avoid them.

**Relationship between Data-Centric Approaches**    As presented in Figure 7, perturb edges, delete nodes and mask attributes (Rong et al., 2019; Trivedi et al., 2022) for graphs are some heuristic ways for data augmentation. The augmented knowledge from them is mainly controlled by human prior knowledge on the perturbations and it often fails to be close to the task, *i.e.,* , random perturbations hardly preserve labels for the augmented graphs. The learning to augment approaches learn from labeled graphs to perturb graph structures (Luo et al., 2022), to estimate graphons for different classes (Han et al., 2022), or to split the latent space for augmentation (Liu et al., 2022a). Although these approaches could preserve labels for the augmented graphs, they introduce less extra knowledge to improve the model prediction. In summary, graph data augmentation is effective in expanding knowledge for limited labels, but it makes no use of unlabeled graphs. Besides,

Figure 7: Qualitative relationship of graphs from different data-centric approach on the task relatedness and contained knowledge.

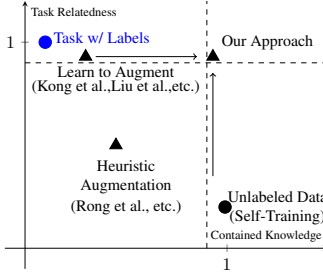

the diversity and richness of the domain knowledge from augmented graphs are far from that contained in a large number of unlabeled graphs. To learn from unlabeled graphs, data-centric approaches like the self-training is assumed to be useful when the unlabeled and labeled data are from the same source. It is less studied when we have a single unified unlabeled source for different tasks.

## B    Additional Method Details

### B.1    Upper bounding the mutual information

In Eq. (6), we use a leave-one-out variant of InfoNCE ($\mathcal{I}_{\text{bound}}$) to derive the upper bound of mutual information. We summarize the derivation (Poole et al., 2019) here.

$$
\begin{aligned}
\mathcal{I}_1(G'; G) &= \mathbb{E}_{p(G,G')}\left[\log \frac{p(G'|G)}{p(G')}\right] \\
&= \mathbb{E}_{p(G,G')}\left[\log \frac{p(G'|G)q(G')}{q(G')p(G')}\right] \\
&= \mathbb{E}_{p(G,G')}\left[\log \frac{p(G'|G)}{q(G')}\right] - \text{KL}(p(G')||q(G')) \\
&\leq \mathbb{E}_{p(G,G')}\left[\log \frac{p(G'|G)}{q(G')}\right]
\end{aligned}
\tag{9}
$$

The intractable upper bound is minimized when the variational approximation $q(G')$ matches the true marginal $p(G')$ (Poole et al., 2019). For each $G_i$, its augmented output $G'_i$, and $M-1$ negative

examples with different labels, we could approximate $q(G'_i) = \frac{1}{M-1} \sum_{j \neq i} p(G'_i | G_j)$. So, we have

$$
\begin{aligned}
\mathcal{I}_1(G'_i, G_i) &\leq \log \frac{p(G'_i | G_i)}{\frac{1}{K-1} \sum_{j=1, j \neq i}^{M} p(G'_i | G_j)} \\
&= \log \frac{p(G'_i | G_i)}{\sum_{j=1, j \neq i}^{M} p(G'_i | G_j)} + \log(M-1) \\
&= \mathcal{I}_{\text{bound}}(G'_i; G_i) + \text{constant}
\end{aligned}
\tag{10}
$$

## B.2    Extraction of Statistical Features on Graphs

For each molecule and polymer graph, we concatenate the following vectors or values for statistical feature extraction.

- the sum of the degree in the graph;
- the vector indicating the distribution of atom types;
- the vector containing the maximum, minimum and mean values of atoms weights in a molecule or polymer;
- the vector containing the maximum, minimum, and mean values of bond valence.

For each protein-protein interaction ego-graph in the biology field, we use the sorted vector of node degree distribution in the graph as the statistical features.

## B.3    Technical Details for Graph Data Augmentation with Diffusion Model

**The Lookup Table from Atom Type to Node Embedding Space**    Given a graph $G$, we assume the node feature matrix on the graph is $\mathbf{X} \in \mathbb{R}^{n \times F_n}$, where $n$ is the number of nodes. The edge feature matrix is $\mathbf{E} \in \mathbb{R}^{m \times F_e}$, where $m$ is the number of edges. There are two ways for $G$ to represent the graph structure in practice. We can use either the dense adjacency matrix $\mathbf{A} \in \mathbb{R}^{n \times n}$ or sparse edge index $\mathbf{I}_e \in \mathbb{R}^{2 \times m}$. The diffusion model (Jo et al., 2022) on graphs prefers the former, which is more straightforward for graph generations. The prediction model prefers the latter because of its flexibility, and less computational cost and time. The transformation between two types of graph structure representation takes additional time. Particularly for molecular graphs, the node features used for generation (one-hot encoding of the atom type) and for prediction (see the official package of OGBG [1] for details) are different, which introduces extra time to process the graph data. For details, we (1) first need to extract discrete node attributes given the atom type and its neighborhoods; (2) we then need to use an embedding table to embed node attributes in a continuous embedding space; (3) the embedding features of nodes with their graph structure are inputted into the graph neural networks to get the latent representation for nodes. The reverse process for data augmentation in DCT may need to repeatedly process graph data with steps (1) and (2). It introduces additional time. To address these technical problems, we build up a lookup table to directly map the atom type to the node embedding. We average the node attributes for the same type of node within the batch. We then use the continuous node attributes as weights to average the corresponding node embedding according to the table.

**Instantiations of SDE on Graphs**    According to Song et al. (2021), we use the Variance Exploding (VE) SDE for the diffusion process. Given the minimal noise $\sigma_{\min}$ and the maximal noise $\sigma_{\max}$, the VE SDE is:

$$
dG = \sigma_{\min} \left( \frac{\sigma_{\max}}{\sigma_{\min}} \right)^t \sqrt{2 \log \frac{\sigma_{\max}}{\sigma_{\min}}} d\mathbf{w}, \quad t \in (0, 1]
\tag{11}
$$

The perturbation kernel is derived (Song et al., 2021) as:

$$
p_{0t}(G^{(t)} \mid G^{(0)}) = \mathcal{N} \left( G^{(t)}; G^{(0)}, \sigma_{\min}^2 \left( \frac{\sigma_{\max}}{\sigma_{\min}} \right)^{2t} \mathbf{I} \right), \quad t \in (0, 1]
\tag{12}
$$

On graphs, we follow Jo et al. (2022) to separate the perturbation of adjacency matrix and node features:

$$
p_{0t}(G^{(t)} \mid G^{(0)}) = p_{0t}(\mathbf{A}^{(t)} \mid \mathbf{A}^{(0)}) p_{0t}(\mathbf{X}^{(t)} \mid \mathbf{X}^{(0)}).
\tag{13}
$$

---

[1] https://github.com/snap-stanford/ogb/blob/master/ogb/utils/features.py

**The Sampling Algorithm in the Reverse Process for Graph Data Augmentation**   We adapt the Predictor-Corrector (PC) samplers for the graph data augmentation in the reverse process. The algorithm is shown in Algorithm 1.

---

**Algorithm 1** Diffusion-Based Graph Augmentation with PC Sampling

---

**Input:** Graph $G$ with node feature $\mathbf{X}$ and adjacency matrix $\mathbf{A}$, the denoising function for node feature $\mathbf{s_X}$ and adjacency matrix $\mathbf{s_A}$, the fine-tune loss $\mathcal{L}_{\textbf{aug}}$, Lagevin MCMC step size $\beta$, scaling coefficient $\epsilon_1$

$\mathbf{A}^{(D)} \leftarrow \mathbf{A} + \mathbf{z}_A; \quad \mathbf{z}_A \sim \mathcal{N}(\mathbf{0}, \mathbf{I})$

$\mathbf{X}^{(D)} \leftarrow \mathbf{X} + \mathbf{z}_X; \quad \mathbf{z}_X \sim \mathcal{N}(\mathbf{0}, \mathbf{I})$

**for** $t = D - 1$ **to** 0 **do**

  $\hat{G}_{(t+1)} \sim p_{0t+1}(\hat{G}_{(t+1)}|G^{(t+1)})$ {inner-loop sampling with another PC sampler}

  $\mathbf{S}_A = \frac{1}{2}\mathbf{s_A}(G^{(t+1)}, t+1) - \frac{1}{2}\alpha\nabla_{\mathbf{A}^{(t)}}\mathcal{L}_{\textbf{aug}}(\hat{G}_{(t+1)})$

  $\mathbf{S}_X = \frac{1}{2}\mathbf{s_X}(G^{(t+1)}, t+1) - \frac{1}{2}\alpha\nabla_{\mathbf{X}^{(t)}}\mathcal{L}_{\textbf{aug}}(\hat{G}_{(t+1)})$

  $\tilde{\mathbf{A}}^{(t)} \leftarrow \mathbf{A}^{(t+1)} + g(t)^2\mathbf{S}_A + g(t)\mathbf{z}_A; \quad \mathbf{z}_A \sim \mathcal{N}(\mathbf{0}, \mathbf{I})$ {Predictor for adjacency matrix}

  $\tilde{\mathbf{X}}^{(t)} \leftarrow \mathbf{X}^{(t+1)} + g(t)^2\mathbf{S}_X + g(t)\mathbf{z}_X; \quad \mathbf{z}_X \sim \mathcal{N}(\mathbf{0}, \mathbf{I})$ {Predictor for node features}

  $\mathbf{A}^{(t)} \leftarrow \tilde{\mathbf{A}}^{(t)} + \frac{\beta}{2}\mathbf{S}_A + \epsilon_1\sqrt{\beta}\mathbf{z}_A; \quad \mathbf{z}_A \sim \mathcal{N}(\mathbf{0}, \mathbf{I})$ {Corrector for adjacency matrix}

  $\mathbf{X}^{(t)} \leftarrow \tilde{\mathbf{X}}^{(t)} + \frac{\beta}{2}\mathbf{S}_X + \epsilon_1\sqrt{\beta}\mathbf{z}_X; \quad \mathbf{z}_X \sim \mathcal{N}(\mathbf{0}, \mathbf{I})$ {Corrector for node features}

**end for**

return $G' = (\mathbf{A}^{(0)}, \mathbf{X}^{(0)})$

---

**The Algorithm of the Framework**   The algorithm of the proposed data-centric knowledge transfer framework is presented in Algorithm 2 and Algorithm 3. In detail, Algorithm 2 corresponds to Section 3.2 and Algorithm 3 corresponds to Section 3.3.

---

**Algorithm 2** The Data-Centric Knowledge Transfer Framework: Learning from Unlabeled Graphs

---

**Input:** Given unlabeled graphs from the space $\mathcal{G}^{[U]}$, randomly initialized score models $\mathbf{s_X}$ and $\mathbf{s_A}$ for node feature and graph adjacency matrix, respectively, the total diffusion time step $T$.

**while** $\mathbf{s_X}$ and $\mathbf{s_A}$ not converged **do**

  Sample $G = (\mathbf{X}, \mathbf{A}) \in \mathcal{G}^{[U]}$

  Sample $t \in \text{Uniform}(1, 2, \ldots, T)$

  Sample $\mathbf{z}_A \sim \mathcal{N}(\mathbf{0}, \mathbf{I})$

  Sample $\mathbf{z}_X \sim \mathcal{N}(\mathbf{0}, \mathbf{I})$

  Sample $\hat{G}$ with $t$, $\mathbf{z}_A$, $\mathbf{z}_X$ and Eq. (13)

  Optimize $\mathbf{s_A}$ with the gradient:

    $\nabla\|\mathbf{z}_A - \mathbf{s_A}(\hat{G}, t)\|^2$

  Optimize $\mathbf{s_X}$ with the gradient:

    $\nabla\|\mathbf{z}_X - \mathbf{s_X}(\hat{G}, t)\|^2$

**end while**

---

**Algorithm 3** The Data-Centric Knowledge Transfer Framework: Generating Task-specific Labeled Graphs

---

**Input:** Given task $k$ with the graph-label space $(\mathcal{G}, \mathcal{Y})$, a randomly initialized prediction model $f_\theta$, the well-trained score model $\mathbf{s} = (\mathbf{s_X}, \mathbf{s_A})$, the training data set $\{G_i, y_i\}_i^{N_t}$, total training epoch $e$, the hyper-paramtere $n$

**for** current epoch $e_i$ from 1 to $e$ **do**

  Train $f_\theta$ on current training data $\{G_i, y_i\}_i^{N_t}$

  **if** $e_i$ is divisible by the augmentation interval **then**

    Select $n$ graph-label pairs with the lowest training loss from $\{G_i, y_i\}_i^{N_t}$

    Get the augmented examples $\{G_i', y_i'\}_i^n$ by Algorithm 1 with the selected examples

    Update $\{G_i, y_i\}_i^{N_t}$ with $\{G_i', y_i'\}_i^n$, *e.g.*, add $\{G_i', y_i'\}_i^n$ to $\{G_i, y_i\}_i^{N_t}$.

  **end if**

**end for**

---

## C   Additional Experiments Set-ups

We perform experiments on 15 datasets, including eight classification and seven regression tasks from chemistry, material science, and biology. We use Area under the ROC curve (AUC) for classification performance and mean absolute error (MAE) for regression.

### C.1   Molecule Classification and Regression Tasks

Seven molecule classification and three molecule regression tasks are from open graph benchmark (Hu et al., 2020). They were originally collected by MoleculeNet (Wu et al., 2018) and used to predict molecule properties. They include (1) inhibition to HIV virus replication in ogbg-HIV, (2) toxicological properties of 617 types in ogbg-ToxCast, (3) toxicity measurements such as nuclear receptors and stress response in ogbg-Tox21, (4) blood–brain barrier permeability in ogbg-BBBP, (5) inhibition to human $\beta$-secretase 1 in ogbg-BACE, (6) FDA approval status or failed clinical trial in ogbg-ClinTox,

(7) having drug side effects of 27 system organ classes in ogbg-SIDER, (8) predicting the property of lipophilicity in ogbg-Lipo, (9) predicting the water solubility ($\log$ solubility in mols per litre) from chemical structures in ogbg-ESOL, (10) predicting the hydration free energy of molecules in water in ogbg-FreeSolv. For all molecule datasets, we use the scaffold splitting procedure as the open graph benchmark adopted (Hu et al., 2020). It attempts to separate structurally different molecules into different subsets, which provides a more realistic estimate of model performance in experiments (Wu et al., 2018).

### C.2 Polymer Regression Tasks

Four polymer regression tasks include GlassTemp, MeltingTemp, ThermCond, and $O_2$Perm. They are used to predict different polymer properties such as *glass transition temperature* ($°C$), *melting temperature* ($°C$), *thermal conductivity* (W/mK) and *oxygen permeability* (Barrer). GlassTemp and MeltingTemp are collected from PolyInfo, which is the largest web-based polymer database (Otsuka et al., 2011). The ThermCond dataset is from molecular dynamics simulation and is an extension from the dataset used in (Ma et al., 2022). The $O_2$Perm dataset is created from the Membrane Society of Australasia portal, consisting of a variety of gas permeability data (Thornton et al., 2012). Since a polymer is built from repeated units, researchers often use a single unit graph with polymerization points as polymer graphs to predict properties. Different from molecular graphs, two polymerization points are two special nodes (see "∗" in Figure 2), indicating the polymerization of monomers (Cormack and Elorza, 2004). For all the polymer tasks, we randomly split by 60%/10%/30% for training, validation, and test.

### C.3 Protein Classification Task

An additional task is protein function prediction using protein-protein interaction graphs (Hu et al., 2019). A node is a protein without attributes, an edge is a relation type between two proteins such as co-expression and co-occurrence. In our DCT, we treat all the relations as the undirected edge without attributes.

### C.4 Baselines and Implementation

When implementing GIN (Xu et al., 2019), we tune its hyper-parameters for different tasks with an early stop on the validation set. We generally implement pre-training baselines following their own setting. For molecule and polymer property prediction and protein function prediction, the pre-trained GIN models with self-supervised tasks such as EDGEPRED, ATTRMASK, CONTEXTPRED in (Hu et al., 2019), INFOMAX (Velickovic et al., 2019) are available. So we directly use them. For other self-supervised methods, we implement their codes with default hyper-parameters. Following their settings, we use 2M ZINC15 (Sterling and Irwin, 2015) to pre-train GIN models for molecule and polymer property prediction. We use 306K unlabeled protein-protein interaction ego-networks (Hu et al., 2019) to pre-train the GIN for the downstream protein function property prediction. For self-training with real unlabeled graphs and INFOGRAPH (Sun et al., 2020), we use 113K QM9 (Ramakrishnan et al., 2014). For self-training with generated unlabeled graphs, we train the diffusion model (Jo et al., 2022) on the real QM9 dataset and then produce the same number of generated unlabeled graphs. To train the diffusion model in our DCT, we also use QM9 (Ramakrishnan et al., 2014).

## D Additional Experiment Analysis

### D.1 The Power of Diffusion Model to Learn from Unlabeled Graphs

In Table 3, when we replace the 133K QM9 with the 249K ZINC (Jo et al., 2022) to train the diffusion model, which nearly doubles the size of the unlabeled graphs and includes more atom types, we do not observe any additional improvement, and in some cases, even worse performance. It is possible because of the constraint of the current diffusion model's capacity to model the data distribution for a much larger number of more complex graphs. It encourages powerful generative models in the future, which could be directly used to benefit predictive models under the proposed framework.

## D.2 Chemical Validity of the Generated Graphs in Downstream Tasks

In Figure 6, we show through some examples that concepts, such as certain chemical rules from the unlabeled graphs, are successfully transferred to downstream tasks. To further validate this point, we gathered 1,000 task-specific graphs generated in the intermediate steps on the tasks of ogbg-BACE, ogbg-BBBP, ogbg-FreeSolv, and $O_2$Perm. We then assessed the chemical validity of these graphs and observed that the validity is 92.8%, 87.9%, 97.4%, and 62.1%, respectively. Results show that transferring knowledge from pre-trained molecular data to target molecules yields relatively high chemical validity. However, the validity drops to 62% when transferring knowledge from pre-trained molecular data to target polymer data. This finding indicates that the transferability of chemical rules becomes more challenging when the distribution gap between the pre-training data and downstream task data is larger.

