# OpenReview forum: "Data-Centric Learning from Unlabeled Graphs with Diffusion Model"
_NeurIPS.cc/2023/Conference — NeurIPS 2023 poster_

### Official Review · Reviewer_3gzg · 2023-06-29

**Soundness:** 2 fair
**Presentation:** 3 good
**Contribution:** 3 good
**Rating:** 5
**Confidence:** 3

**Summary:**

The paper discusses the problem of graph property prediction tasks, where there is a limited amount of labeled data and a large set of unlabeled graphs available. The conventional approach is to train a model using self-supervised tasks on the unlabeled graphs and then fine-tune it for prediction tasks. However, the knowledge gained from self-supervised tasks may not align with the requirements of the prediction tasks.

To address this issue, the authors propose a data-centric approach that extracts knowledge from the unlabeled graphs to enhance each property prediction model. They introduce a diffusion model to utilize the unlabeled graphs fully and two new objectives that guide the model's denoising process and generate task-specific graph examples and labels. Specifically, the first objective minimizes an upper bound of mutual information between the augmented and the original graphs in the graph space. The second objective maximizes the probability of the predicted label of augmented graphs being the same as the label of original graphs.

Experimental results demonstrate that the proposed data-centric approach outperforms fifteen existing methods on various graph property prediction tasks. The improvement in performance is attributed to the generated labeled examples, which are more effective than self-supervised learning. The authors also discuss related work in graph property prediction and learning from unlabeled data, highlighting the limitations of existing methods and the need for a data-centric approach. They present their framework (DCT) in detail and provide results showing its superior performance compared to baseline methods used for comparison.

**Strengths:**

1. The proposed method offers a novel approach to graph property prediction by employing a diffusion model, setting it apart from traditional techniques.

2. Empirically, the introduced method surpasses other techniques in various areas including self-supervised learning, semi-supervised learning, and graph data augmentation.

**Weaknesses:**

1. The discussion of the graph diffusion model is insufficient. The proposed method relies on the graph diffusion model as its backbone technique. A clear discussion of the similarities and differences between this work and other graph diffusion model methods [1,2] can clarify the contribution of this work.

2. The interpretability of "visible" knowledge transfer from unlabeled graphs is interesting, but only several case studies cannot support the claim that "concepts such as some chemical rules from the unlabeled graphs are successfully transferred to downstream tasks" (lines 318-319).



[1] Jo, Jaehyeong, Seul Lee, and Sung Ju Hwang. "Score-based generative modeling of graphs via the system of stochastic differential equations." In *International Conference on Machine Learning*, pp. 10362-10383. PMLR, 2022.

[2] Vignac, Clement, Igor Krawczuk, Antoine Siraudin, Bohan Wang, Volkan Cevher, and Pascal Frossard. "DiGress: Discrete Denoising diffusion for graph generation." *arXiv preprint arXiv:2209.14734* (2022).

**Questions:**

1. In lines 46-48, the generation of labeled examples is part of the property prediction task pipeline. Please describe how to use these generated task-specific labeled examples to bridge the gap between the final prediction and the generated graph.

2. In lines 60-61, it is described that the trained predictor model f is used in the first objective. How is the predictor trained? If it is trained on a small amount of labeled data, then why is the not-well-trained predictor (as stated in Line 119) able to help with graph generation?

3. Can other graph diffusion model methods be applied to the graph property prediction task with trivial adaptation? If not, why?

---

> ### Author Rebuttal · Authors · 2023-08-07
>
> ### To weakness 1 about the literature review on graph diffusion model
>
> Thanks for your comments. Here are discussions on recent advances for diffusion models on graphs.
>
> Recent works have improved the generative modeling of diffusion models on graphs [1,2,3,4,5]. EDP-GNN [1] employed score matching for permutation-invariant graph data distribution. GDSS [2] extended continuous-time framework [6] to model node-edge joint distribution. DiGress [3] used the transition matrix to preserve the discrete natures of the graph structure. GraphARM [4] introduced a node-absorbing autoregressive diffusion process. EDGE [5] focused on efficiently generating larger graphs. Instead of improving the generation capacity of the diffusion model, our model builds on [2,6] for predictive tasks, i.e., graph classification and graph regression. Our focus is on improving the transfer of knowledge from unlabeled graphs to various downstream tasks with the diffusion model.
>
> We will add the above discussion to the main text.
>
> ### To weakness 2 about lines 318-319
>
> Thank you for your valuable suggestions. To validate our hypothesis regarding the transfer of chemical rules from unlabeled graphs to downstream tasks, we gathered 1,000 task-specific graphs generated in the intermediate steps. We then assessed the chemical validity of these graphs on two classification tasks (ogbg-BACE and ogbg-BBBP) and two regression tasks (ogbg-FreeSolv and O$_2$Perm). The outcomes of these evaluations are summarized in the table below.
>
> |              |  BACE   | BBBP    | FreeSolv | O$_2$Perm |
> |--------------|---------|---------|----------|--------|
> | Validity (\%) | 0.928   | 0.879   | 0.974    | 0.621  |
>
> First, we observe that the validity is close to 90\% in most cases, validating our claim. Furthermore, transferring knowledge from pre-trained molecular data to target molecules has relatively high chemical validity. However, the validity drops to 62\% when transferring knowledge from pre-trained molecular data to target polymer data. This finding indicates that the transferability of chemical rules becomes more challenging when the distribution gap between pre-training data and downstream task data is larger. We will incorporate these findings into the main text accordingly.
>
> ### To question 1 about lines 46-48
>
> Thanks for your suggestions. We described our high-level ideas in lines 43--49. Technical details about the implementation of our idea could be found in lines 50-65 and all of Section 4. Specifically for your question for lines 46-48, we break the question into two parts: (1) how to generate graphs to close the gap between the unlabeled graphs and a specific task (2) how to use the generated task-specific graphs. A brief discussion on (1) could be found in lines 56-64 and details could be found in Section 4.3 (Lines 169-211). A brief discussion on (2) could be found in lines 64-65 and details could be found in Section 4.3 (Lines 212--226)
>
> Here we summarize the important points. For (1), we use two task-related losses to guide the generation of the diffusion model. So, the generated graphs are task-specific. For (2), we use an iterative process to alternatively generate task-specific graph data and train the downstream prediction model. The model is first trained on all the downstream labeled data and then the partially trained model is used to guide the generation of task-specific generated graphs. Then we start a new iteration and repeat training the model.
>
> ### To question 2 about lines 60-61
>
> Thanks for your question, as stated in lines 64-65:
>
> > DCT iteratively generates new examples to augment the labeled dataset and progressively trains the prediction model with it.
> >
>
> We will iteratively update the downstream prediction model ($f$). We hypothesize that model training and task-specific data generation mutually enhance each other. This assumption is common in self-training [7,8,9,10]. We also empirically validated this assumption in Figure 5. Details are in lines 305-311.
>
> ### To question 3 about the use of other diffusion models
>
> We adopt the general framework [2,6] for continuous-time diffusion modeling via stochastic differential equations (SDE). Practical implementation requires discretizing SDE in both the diffusion and reverse process, yielding various model instances like SMLD [11] and DDPM [12]. Our framework could accommodate these diffusion models and their extensions with minimal adjustments. We note that the framework [2,6] uses Gaussian noise for data perturbation and noise estimation in reverse processes, while DiGress [3] employs discrete noise, predicting clean graphs from perturbed ones. Corresponding code adjustments are needed for different noise types and model outputs.
>
> ### Reference
>
> [1] Permutation invariant graph generation via score-based generative modeling. AISTATS, 2020.
>
> [2] Score-based generative modeling of graphs via the system of stochastic differential equations. ICML 2022.
>
> [3] Digress: Discrete denoising diffusion for graph generation. ICLR 2023.
>
> [4] Autoregressive Diffusion Model for Graph Generation. ICML 2023.
>
> [5] Efficient and Degree-Guided Graph Generation via Discrete Diffusion Modeling. ICML 2023.
>
> [6] Score-based generative modeling through stochastic differential equations. ICLR 2021.
>
> [7] Label propagation for deep semi-supervised learning. CVPR 2019.
>
> [8] Pseudo-label: The simple and efficient semi-supervised learning method for deep neural networks. Workshop at ICML 2013.
>
> [9] Crest: A class-rebalancing self-training framework for imbalanced semi-supervised learning. CVPR 2021.
>
> [10] Multi-task self-training for learning general representations. CVPR 2021.
>
> [11] Generative modeling by estimating gradients of the data distribution. NeurIPS 2019.
>
> [12] Denoising diffusion probabilistic models. NeurIPS 2020.

---

> > ### Comment · Reviewer_3gzg · 2023-08-20
> > **Official Comment by Reviewer 3gzg**
> >
> > Having received and reviewed the authors' responses to my queries, I'd like to amend my original score from 4 to 5 for this submission.

---

> > > ### Author Response · Authors · 2023-08-21
> > >
> > > We are delighted to address all your concerns. Thanks again for your valuable comments and for raising the score.

---

### Official Review · Reviewer_trSH · 2023-07-06

**Soundness:** 3 good
**Presentation:** 4 excellent
**Contribution:** 3 good
**Rating:** 6
**Confidence:** 3

**Summary:**

This paper proposes a new data augmentation method called DCT based on unlabeled graphs. The proposed DCT generates new augmented graphs using a diffusion model from unlabelled graphs and also with labeled graphs from down-stream tasks as guidance. Two specific objectives are proposed to guide the model's denoting process to generate task-specific graph examples and their corresponding labels. Experimental results demonstrate the superior performance of the proposed method.

**Strengths:**

1. A new graph augmentation method is proposed based on diffusion models developed in the computer vision field. The exploration of diffusion models on graphs is relatively new.
2. The proposed method adopts two task-specific objectives, enabling to learn task-specific knowledge and generate task-related labeled graphs. This mechanism has been shown effective in addressing the shortcomings of self-supervised learning self-training methods.
3. The experimental evaluation and ablation studies demonstrate the promising performance of the proposed method on graph property prediction tasks.


**Weaknesses:**

1. The idea of using diffusion models to generate graphs is borrowed from the field of computer vision, where the diffusion and reverse process is more intuitive. However, it remains unclear how it can be applied to both graph structure and node features.
2. Some relevant literature review, e.g., on pseudo-labelling, is missing in related work.



**Questions:**

1. Does your method suffer from overfitting? In Figure 4, the performance of the proposed method seems to decrease as the number of selected labeled graphs for data augmentation increases. Although it is beneficial to consider downstream tasks when augmenting graphs, the potential overfitting problem should be discussed.
2. How is the diffusion and reverse process applied to both graph structure and node features? Do you perturb different types of noise to graph structure/node features? More detailed discussions are necessary.
3. In Definition 4.2, how to define \bar{G}?

**Limitations:**

The authors have not discussed potential limitations including societal impact of their work.

---

> ### Author Rebuttal · Authors · 2023-08-07
>
> Thank you for your feedback model details and literature review. We offer detailed responses to each of your points.
>
> ### To weakness 1 and question 2 about the diffusion model on graphs
>
> We follow [1,2] to define graph diffusion models. [2] unified score-based generative models and denoising diffusion probabilistic models, using stochastic differential equations (SDE) in continuous state spaces. [1] extended [2] to graphs. Besides [2], there are many recent work that has verified the effectiveness of the diffusion models on graphs [3,4,5]. We also want to point out that, differing from the existing work [1,3,4,5], which mainly focused on advancing the generative performance of the diffusion model, we propose to leverage the diffusion model for predictive tasks via data augmentation and knowledge transfer. Here, we present a more careful review to discuss recent advances for diffusion models on graphs.
>
> Given a graph data point $G \in \mathcal{G}$ in the graph space with the adjacency matrix $\mathbf{A} \in \mathbb{R}^{N \times N}$ and the node feature matrix $\mathbf{X} \in \mathbb{R}^{N\times F}$, where $N$ is the number of nodes and $F$ is the dimension of node features, [1] extended the above SDE to the graph data $G \in \mathbb{R}^{N\times N} \times \mathbb{R}^{N\times F} $ (see our Eq. (3) and (4) in lines 152-167). Following [1], we perform functions on graphs, such as $\mathbf{f}(\cdot,t)$ in Eq. (3) and (4), separately on the node feature variable $\mathbf{X}$ and adjacency matrix variable $\mathbf{A}$. The marginal distribution of the graph probability $p_t(G^{(t)})$ at time step $t$ is also defined from two aspects: the node feature $\mathbf{X}$ and the adjacency matrix $\mathbf{A}$. The ground-truth gradient for the probability or $\log p_t(G^{(t)})$ is estimated by two graph neural networks: one estimates $\nabla_{\mathbf{X}^{(t)}} \log p_t (\mathbf{X}^{(t)}, \mathbf{A}^{(t)})$ for node features, and the other estimates $\nabla_{\mathbf{A}^{(t)}} \log p_t (\mathbf{X}^{(t)}, \mathbf{A}^{(t)})$ for adjacency matrices. These neural networks are trained by score matching methods [2]. We kindly note that certain descriptions are available in our paper, particularly in Section 4.2. Unfortunately, due to the page limit, many details were moved to the appendix such as the B.3 section. For more details, including equations, instantiations, and algorithms, we kindly direct reviewers to the appendix. Additionally, we will add more details to the main text.
>
> ### To weakness 2 about the literature review on pseudo-labeling
>
> Thanks for your suggestion. Pseudo-labeling is a common technique for semi-supervised learning [6,7]. It first trains a model that iteratively assigns pseudo-labels to the set of unlabeled training examples. The pseudo-labeled examples are then used to enrich the labeled training set. And the model continues training with the updated training set. Many studies have explored improving uncertainty estimation [8,9,10] to help the model filter out noise for reliable pseudo-labels. Recently, pseudo-labels have been applied in imbalanced learning [11] and representation learning [12]. Pseudo-label methods are restricted to confidently predictable labels and may ignore the huge number of other unlabeled data.
>
> ### To question 1 about performance drop in Figure 4 when increasing top-$n$
>
> The performance drop from increasing augmented graphs per iteration could have multiple reasons. A key factor is the noisy graphs introduced with a high hyper-parameter $n$ in Figure 4 (e.g., > 30\%). We use an iterative process for training and augmented graph creation, expecting mutual enhancement (confirmed in Figure 5). As the downstream prediction model isn't perfect during augmented graph creation, we select only top-$n$ lowest property prediction loss to prevent noisy graph generation. When the top-$n$ is very high, like larger than 30\%, the noisy graphs would cause the performance drop as shown in Figure 4. We have discussed this phenomenon in lines 301-304 and lines 224-227.
>
> Regarding the concern of overfitting, we have briefly talked about it in lines 178-183. We discussed the balance between task-relatedness and diversity in generating augmented graphs. Limited diversity can lead to overfitting, thus we employ minimality (definition 4.2) to enhance augmented graph diversity.
>
> ### To question 3 about the explanation of Definition 4.2
>
> Thank you for your comments. Here we provide more details.$\forall \bar{G}$ represents any augmented graph that sufficiently preserves the original graph's label (as defined in Definition 4.1). We will add these details to clarify this definition in the paper.
>
> ### Reference
>
> [1] Score-based generative modeling of graphs via the system of stochastic differential equations. ICML 2022.
>
> [2] Score-based generative modeling through stochastic differential equations. ICLR, 2021.
>
> [3] Autoregressive Diffusion Model for Graph Generation. ICML 2023.
>
> [4] Efficient and Degree-Guided Graph Generation via Discrete Diffusion Modeling. 2023 ICML.
>
> [5] Digress: Discrete denoising diffusion for graph generation. ICLR 2023.
>
> [6] Label propagation for deep semi-supervised learning. CVPR 2019.
>
> [7] Pseudo-label: The simple and efficient semi-supervised learning method for deep neural networks. Workshop at ICML 2013.
>
> [8] Dropout as a bayesian approximation: Representing model uncertainty in deep learning. ICML 2016.
>
> [9] Single-model uncertainties for deep learning. NeurIPS 2019.
>
> [10] Deep evidential regression. NeurIPS 2020.
>
> [11] Crest: A class-rebalancing self-training framework for imbalanced semi-supervised learning. CVPR 2021.
>
> [12] Multi-task self-training for learning general representations. CVPR 2021.

---

### Official Review · Reviewer_T9FT · 2023-07-07

**Soundness:** 3 good
**Presentation:** 3 good
**Contribution:** 2 fair
**Rating:** 6
**Confidence:** 5

**Summary:**

The paper leverages diffusion SDE to approach the hidden chemical space and help the prediction tasks of molecular properties.

**Strengths:**

1. It is a good idea to leverage mutual information for the alignment of the hidden chemical space within both labeled and unlabeled graphs.
2. The results presented in this study seem to exhibit superior performance when compared to other methods examined in the paper, which is an encouraging observation.

**Weaknesses:**

1. The paper could benefit from greater originality. The primary concepts echo closely those found in references [1] and [2]. For instance, [1] utilized generative models to boost the prediction performance for an array of graph datasets, while [2] applied a score matching method to uncover the hidden chemical space, thereby facilitating active learning in the predictive task of molecular properties. I recommend an expanded discussion within the paper to clearly delineate the distinctions between the current study and the methods in references [1] and [2]. This would more effectively highlight the unique contributions of this work.

2. The paper appears to have an insufficiency in its literature review section. It would greatly benefit from an evaluation of self-supervised learning-based molecular pre-training models as discussed in references [3], [4], and [5]. These studies could provide significant context and further inform the study's approach.

3. The paper reports an incremental advancement in performance. Upon reviewing the OGB leaderboard [6], the results presented for ogbg-molhiv do not appear to supersede those achieved by SOTA methods currently leading on the leaderboard. Moreover, it seems that the performance does not even align favorably with the results presented in references [3], [4], and [5]. The authors are encouraged to provide an explanation or delve into the possible reasons behind this observed discrepancy in performance levels.

[1] Liu, Songtao, et al. "Local augmentation for graph neural networks." International Conference on Machine Learning. PMLR, 2022.

[2] Zhou, Kuangqi, et al. "Jointly Modelling Uncertainty and Diversity for Active Molecular Property Prediction." Learning on Graphs Conference. PMLR, 2022.

[3] Zhang, Zaixi, et al. "Motif-based graph self-supervised learning for molecular property prediction." Advances in Neural Information Processing Systems 34 (2021): 15870-15882.

[4] Rong, Yu, et al. "Self-supervised graph transformer on large-scale molecular data." Advances in Neural Information Processing Systems 33 (2020): 12559-12571.

[5] Xu, Minghao, et al. "Self-supervised graph-level representation learning with local and global structure." International Conference on Machine Learning. PMLR, 2021.

[6] https://ogb.stanford.edu/docs/leader_graphprop/#ogbg-molhiv

**Questions:**

See above

---

> ### Author Rebuttal · Authors · 2023-08-07
>
> We sincerely value your feedback. We respectfully and firmly defend the originality, the literature review, and a fair evaluation of the performance of our model. We begin by offering a concise overview of our rebuttal, followed by a comprehensive point-by-point elaboration.
>
> ### Summary
> ($\text{The first six references strictly adhere to the order used by the reviewer}$.)
>
> 1. Originality: Our novelties and motivations in lines 43-49: transfer learning by data augmentation, are essentially different from the reference [1,2]. Another clear difference is that we focused on graph property prediction with automatic knowledge transfer from unlabeled graphs, while [1] focused on node classification and [2] focused on active learning.
>
> 2. Literature review: **We have already compared both methods** from the reference [3] (MGSSL) and from the reference [5] (GraphLoG) in our reported results in Table 1 (the 6th and 7th lines in the self-supervised part) and **particularly reviewed the disadvantages** of MGSSL [3] in lines 28-33. The reference [4] is found not as good as the MGSSL [3] in the study of the reference [3].
>
> 3. Model performance: To ensure unbiased conclusions, we need (and have conducted) a fair evaluation, which should exclude complex training techniques like domain knowledge fingerprints and hybrid models. These techniques are broadly used in the ogbg-HIV leaderboard.
>
> We give a detailed response to support our summary as follows.
>
> ### To question 1 about originality:
>
> Our major novelties: transfer learning by data augmentation is essentially different from the reference [1,2].
>
> We focus graph-level knowledge transfer, whereas [1] concentrates more on node-level tasks where the local graph structure holds greater importance. As indicated in the reported performance from [1] on the graph-level task ogbg-HIV, which stands at 76.18, it appears to be lower than our GIN implementation which achieves 77.4. These results highlight the distinctions between node-level and graph-level approaches. For example, we find that local structures like aromatic rings could be misleading (see Lines 28-33 in our paper). Therefore, the gap between unlabeled distribution and task label distribution for graph data objectives is more important to address in our task.
>
> The work [2] focused on: active learning for molecular property prediction. Active learning needs an oracle such as chemists, while we focus on automatically discovering useful knowledge from massive unlabeled graphs and transferring them to downstream tasks.
>
> In summary, the work [1,2] is interesting. We use generative modeling techniques in common, but it is crucial and undeniable to emphasize that our main focus lies in different areas, as described earlier. We will add extended discussions for potential connections and applications about node-level tasks and active learning. However, as demonstrated above, our originality clearly distinguishes it from [1,2].
>
> ### To question 2 about the lack of literature review and evaluation
>
> We would like to politely clarify that \textbf{we have already compared} both methods (MGSSL) [3] and (GraphLoG) [5] and particularly **reviewed** the disadvantages of MGSSL [3]. You can find our discussion on MGSSL [3] in lines 27-33 and a comparison with [3] and [5] in Table 1 (the 6th and 7th lines in the self-supervised part). The work [4] was proposed in 2020 and has been beaten by MGSSL, which was proposed in 2021. We would like to prefer more advanced techniques for comparison to leave more page space to comprehensively study the effectiveness of the proposed framework. However, we remain open to expanding our discussion for the reference [4], if it holds the potential to yield a broader impact on our research.
>
> ### To question 3 about OGBG leaderboard
>
> It is worth noting that the top-performing approaches on the ogbg-hiv leaderboard are the result of combining various training tricks and incorporating domain knowledge. Here are a few examples: (1) Fingerprint. It consists of domain knowledge rule extraction with expertise knowledge; (2) Advanced graph neural network (GNN) structures; (3) Hybrid models from different machine learning models and GNN models. Most of the existing graph pre-training models [5,7,9] do not include these tricks. We want to evaluate the model in line with these pre-training baselines. Besides, developing our method following the simplest setting helps avoid drawing spurious conclusions attributed to more complex models. In a fair evaluation, our method demonstrates notable improvements compared to 15 state-of-the-art baselines from self-supervised learning, semi-supervised learning, and graph data augmentation. Specifically, we achieve a reduction of 13.4\% in mean absolute error for the molecule graph regression task and 10.2\% for the polymer graph regression task.
>
> ### To question 3 about the reported performance in [3,4,5]
>
> We want to emphasize that direct comparisons between the reported numbers in our paper and those in papers [3,4,5] are not fair. The reason for the difference in reported numbers is due to different data splitting and underlying prediction models used in the evaluation process. While we generally follow the scaffold splitting method, the results may differ with different random seeds [7,8]. Additionally, in the work [4], three random-seeded scaffold splittings were used with the graph transformer as the backbone. These differences make direct comparisons with reported numbers in different papers impossible. In our paper, we ensure a unified and fair evaluation setting by implementing all baselines following [8].
>
> ### Reference
>
> ($\text{The first six references strictly adhere to the order used by the reviewer}$.)
>
> [7] Strategies for pre-training graph neural networks. ICLR 2020.
>
> [8] Open graph benchmark: Datasets for machine learning on graphs. NeurIPS 2020.
>
> [9] Graph self-supervised learning with accurate discrepancy learning. NeurIPS 2022.

---

> > ### Comment · Reviewer_T9FT · 2023-08-15
> > **The comparison to SOTA methods are missing**
> >
> > I appreciate the author's detailed response, particularly where answer 1 and 3 have addressed the concerns I raised. Nonetheless, I harbor reservations regarding the 2nd answer. Firstly, the work from NeurIPS 2020 did not surpass the SOTA. The author also made comparisons with two studies published in 2019, i.e. Hu et al., 2019 and Velickovic et al., 2019. Secondly, each experiment has its own distinct settings and datasets. Thus, the better performance of  MGSSL than GROVER[4] reported in MGSSL paper does not obviate the necessity for comparative analysis in alternative settings or datasets. The apparent lack of specific SOTA methods in this comparative experiment prompts the question: Are there substantial differences between the proposed method and the omitted method that lead to cherry-picking of the compared methods? In light of this, I strongly advocate more exhaustive comparisons with all existing SOTA methods in experimental evaluations.

---

> > > ### Author Response · Authors · 2023-08-15
> > >
> > > We value your feedback and are pleased to have addressed your questions 1 and 3 regarding the novelty and performance of our method.
> > >
> > > We respectfully disagree with your comment: "The apparent lack of specific SOTA methods in this comparative experiment" and also disagree with your previous comment in the second question: "The paper appears to have an insufficiency in its literature review" **These assertions may overlook several important points covered in our paper.**
> > >
> > > Three papers [3, 4, 5] are referenced as evidence for your comments: GROVER [4] was presented at NeurIPS 2020, MGSSL [3] at NeurIPS 2021, and GraphLoG [5] at ICML 2021. **We would like to emphasize once again that we have already included the comparison with MGSSL [3] and GraphLoG [5] in Table 1.** As previously mentioned, we initially considered MGSSL [3] rather than GROVER [4] because both GROVER [4] and MGSSL [3] utilized graph structures (motifs) for self-supervised tasks, and MGSSL [3] demonstrated better performance over GROVER [4] in fair and direct comparisons. We chose the **better one** from MGSSL [3] and GROVER [4] to allocate space for other valuable studies within the main text. Therefore, the assertion: "cherry-picking of the compared methods" is unjust. Here, we provide empirical validation through additional experiments. The table below displays the performance of the pre-trained GROVER [4] base model, fine-tuned on classification tasks (OGBG-BBBP and OGBG-BACE) and regression tasks (OGBG-FreeSolv and OGBG-ESOL) over three runs. We follow the default settings in the official implementation.
> > >
> > > |                              | BBBP  &uarr;    | BACE  &uarr;    | FreeSolv  &darr;   | ESOL  &darr;     |
> > > |------------------------------|-----------|-----------|--------------|--------------|
> > > | GIN                          | 67.5(2.7) | 77.5(2.8) | 1.639(0.146) | 0.766(0.016) |
> > > | GROVER                       | 63.3(0.1) | 78.4(3.1) | 2.539(0.523) | 1.263(0.125) |
> > > | Best graph self-supervised   | 69.9(0.5) | 81.3(2.4) | 1.952(0.088) | 0.935(0.018) |
> > > | Best graph semi-supervised   | 66.7(1.9) | 78.4(3.0) | 1.547(0.082) | 0.724(0.082) |
> > > | Best graph data augmentation | 70.2(1.0) | 82.4(2.4) | 1.565(0.098) | 0.755(0.039) |
> > > | Ours                         | 70.8(0.5) | 85.6(0.6) | 1.339(0.075) | 0.717(0.020) |
> > >
> > > Results support our point that: the omitted method **does not** lead to cherry-picking of the compared methods. Our results also show that semi-supervised learning and graph data augmentation methods are competitive and sometimes even outperform the state-of-the-art self-supervised learning approaches. We have discussed the potential reasons for the limitations of self-supervised learning in molecular property prediction tasks in our motivation (lines 20-42) and presented experiment analysis in lines 256-276. Our results align closely with recent observations [13].
> > >
> > > Besides, the assertion that "The comparison to SOTA methods is missing" is potentially misleading. **We selected 15 baselines from graph self-supervised learning, graph semi-supervised learning, and graph data augmentation methods** to comprehensively cover a wide range of representative and state-of-the-art approaches. Our baselines include pioneering methods like those (Hu et al., 2019 and Velickovic et al., 2019) as mentioned by you, as well as the state-of-the-art methods in 2022 such as D-SLA [10], G-Mixup [11], and GREA [12]. We believe that our comparison with recent and broader approaches would reflect the advancements and trends in the field of molecular property prediction.
> > >
> > > ### Reference
> > >
> > > [10] Graph self-supervised learning with accurate discrepancy learning. NeurIPS 2022.
> > >
> > > [11] G-Mixup: Graph Data Augmentation for Graph Classification. ICML 2022.
> > >
> > > [12] Graph Rationalization with Environment-based Augmentations. KDD 2022.
> > >
> > > [13] Does gnn pretraining help molecular representation? NeurIPS 2022.

---

> > > > ### Comment · Reviewer_T9FT · 2023-08-16
> > > > **Good experimental reports**
> > > >
> > > > I am grateful for the author's clarifications. Right now, the results looks reasonable to me. All of my concerns have been addressed. I will raise my score accordingly. I would also suggest the authors to incorporate the content of the rebuttal into the main body of the paper to enhance it.

---

> > > > > ### Author Response · Authors · 2023-08-16
> > > > > **Thank you for raising the score**
> > > > >
> > > > > We are delighted to have addressed all of your concerns. Thank you for raising the score. We will incorporate our discussions into further improving the paper. Thank you for your valuable feedback.

---

> > ### Comment · Reviewer_T9FT · 2023-08-15
> >
> >  By the way, just curious. Given that the Diffusion Model is trained on unlabeled data, would it be feasible to employ this model for the generation of novel molecules?

---

> > > ### Author Response · Authors · 2023-08-15
> > >
> > > Thank you for your interesting questions. As visualized in Figure 6, the diffusion model can generate desirable molecules with the guidance of the downstream predictor and targets. We believe it would be a promising direction to use our framework for the generation of novel molecules. We leave it for future work and focused on the prediction tasks in this paper.

---

### Official Review · Reviewer_6AKL · 2023-07-07

**Soundness:** 3 good
**Presentation:** 2 fair
**Contribution:** 3 good
**Rating:** 6
**Confidence:** 4

**Summary:**

This paper studies leveraging unlabeled graphs to aid the supervised graph learning tasks with limited labeled graphs. This paper starts with a motivation that previous self-supervised methods may conflict with the downstream task and lead to poor performance. This paper proposes to model the graph distribution with diffusion models and train the diffusion model jointly with the downstream prediction model. The proposed method optimizes the prediction loss of the augmented graphs and maximizes the difference between the augmented graph and the original labeled graph. This paper evaluates the proposed method on 15 graph property prediction tasks, including seven molecule classification, three molecule regression tasks, four polymer regression tasks, and one protein function prediction task. Compared to previous self-supervised learning, semi-supervised learning, and data augmentation approaches, the proposed method shows significant improvement over previous methods.

**Strengths:**

- This paper leverage diffusion models to learn the distribution of unlabeled graph and use the model to augment labeled graph for downstream tasks. In contrast to previous self-supervised learning methods, the proposed method is more effective in terms of adapting to each downstream task.
- Empirical evaluations are extensive. This paper provides a broad range of experiments that demonstrate the advantage of the proposed method over 15 previous methods on various datasets. This paper also presents ablation studies and case studies that show the working of the proposed method.

**Weaknesses:**

Details of the proposed method need to be clarified, such as the training of the diffusion model and implementation details. Please see Questions for more details.

**Questions:**

- The proposed method uses a diffusion model to first learn graph distribution from unlabeled graphs. Is the diffusion model trained for each task? Or is a joint diffusion model trained and applied to every task?
- What dataset is the diffusion model trained on in the experiments?
- Is the prediction model trained on both originally labeled graphs and augmented graphs in the downstream tasks? How would the performance be if the method also trained on the original labeled graphs?
- What would be the perturbation scale between originally labeled graphs and augmented graphs, which leads to the best performance?

**Limitations:**

This work has discussed the limitations of the proposed method.

---

> ### Author Rebuttal · Authors · 2023-08-07
>
> Thank you for your feedback on the model details. Below, we offer detailed responses to each of your points.
>
> ### To question 1 and 2 about the training of the diffusion model and dataset
>
> We train one diffusion model for 14 molecule and polymer tasks on the unlabeled QM9 datasets.
>
> We consider another task on protein-protein interaction (PPI) graphs. These graphs have proteins as nodes and protein-protein relations as edges, whereas molecular graphs have atoms as nodes and bonds as edges. It is not reasonable to transfer the knowledge about molecular graph structure to protein-protein interaction structure. Therefore, another diffusion model is trained on unlabeled PPI graphs for the downstream PPI tasks.
>
> More details could be found in the baselines and implementation paragraph (lines 240-252).
>
> ### To question 3 about the training of the downstream prediction model
>
> We adopt an iterative approach to train the downstream prediction model and create augmented graphs. As a result, the prediction model is trained on both the original labeled graphs and the newly generated labeled graphs from the second iteration onwards. More details are in lines 212-227. The GIN model serves as the backbone, and we also report its performance in Table 1 as it is trained solely on the original labeled graphs.
>
> ### To question 4 about the perturbation
>
> Thank you for your insightful questions. We use the hyper-parameter, the perturbation step $D$, to represent the scale of perturbation applied to the augmented graphs. To analyze the sensitivity of this hyper-parameter, we conducted experiments with $D$ ranging from 1 to 10, as illustrated in the top figure of Figure 4. We observed that DCT remains robust across a broad range of $D$ values, and we recommend setting $D$ to 5 as a default value. Further details on this analysis can be found in Section 5.3, specifically in lines 300 to 301 of the paper.

---

> > ### Comment · Reviewer_6AKL · 2023-08-12
> > **Thanks for the response!**
> >
> > I have read the authors' responses. The responses have addressed my concerns. Therefore, I will keep my score.

---

> > > ### Author Response · Authors · 2023-08-12
> > > **Thank you for your reply**
> > >
> > > We highly appreciate your valuable time to review our paper. It is encouraging to know that we have effectively addressed all your concerns and questions. We remain receptive to any further questions, discussions, or suggestions you might have regarding our work.

---

### Author Response · Authors · 2023-08-19
**Summary of discussion**

Dear reviewers,

We appreciate all the valuable feedback from four different aspects: novelty, literature review, method detail, and experiment. To facilitate the discussion, we have prepared a summary for your consideration. Here we outline how we have addressed the concerns raised by each reviewer.

### Reviewer 6AKL

1. Method detail: We explained the training details of the diffusion model.

2. Experiment: We provided a detailed explanation of the dataset used in our experiments. We also analyzed the impact of the perturbation scale on model performance.

### Reviewer T9FT

1. Novelty: We have highlighted a major novelty: knowledge transfer from unlabeled graphs by data augmentation. This sets our work apart from the two referenced papers as mentioned by the reviewer, which focused on different tasks - one on node-level tasks and the other on active learning.

2. Literature review: We showed that we have covered two referenced papers as mentioned by the reviewer. We also showed that the literature review on baselines is broad and sufficient.

2. Experiment: We have addressed concerns about the performance and introduced experiments comparing a new baseline, demonstrating that our baseline selection and comparison do not lead to cherry-picking presentation.

### Reviewer trSH

1. Literature review: We discussed more work related to pseudo-labeling.

2. Method detail: We explained the details of using diffusion for graph generation tasks. We also clarified the definition of $\bar{G}$.

3. Experiment: We explained that the augmented examples can introduce noise as the top-$n$ hyper-parameter increases, potentially causing a performance drop in Figure 4.

### Reviewer 3gzg

1. Literature review: We discussed more work related to the diffusion model. We also discussed how the proposed framework can adapt to various diffusion models.

2. Method detail: We explained how to generate desirable graphs in downstream tasks and how to use them with the downstream predictor model.

3. Experiment: We presented more evidence to support our claim on the visible knowledge transfer.


We are encouraged by successfully addressing the concerns raised by Reviewer 6AKL and T9FT through discussions. We sincerely apologize if we missed any important concerns that the reviewers believe were not adequately summarized or addressed. We respectfully request the opportunity to further address these concerns during the remaining discussion period. Your feedback is highly valued.

---

### Decision · Program_Chairs · 2023-09-21

**Decision:**

Accept (poster)

**Comment:**

This paper leverage diffusion models to learn the distribution of unlabeled graph and use the model to augment labeled graph for downstream tasks. The reviewers acknowledged the completeness of the extensive experimental study, although showed slight concerns about originality and a more thorough literature review.